# An adhesion G protein-coupled receptor is required in cartilaginous and dense connective tissues to maintain spine alignment

Zhaoyang Liu[1,2], Amro A Hussien[3,4], Yunjia Wang[1,5], Terry Heckmann[1], Roberto Gonzalez[1], Courtney M Karner[6], Jess G Snedeker[3,4], Ryan S Gray[1,2]*

[1]Department of Pediatrics, Dell Pediatric Research Institute, 1400 Barbara Jordan Blvd, The University of Texas at Austin, Dell Medical School, Austin, United States; [2]Department of Nutritional Sciences, The University of Texas at Austin, Austin, United States; [3]Department of Orthopedics, Balgrist University Hospital, University of Zurich, Zurich, Switzerland; [4]Institute for Biomechanics, ETH Zurich, Zurich, Switzerland; [5]Department of Spine Surgery and Orthopaedics, Xiangya Hospital, Central South University, Changsha, China; [6]Department of Internal Medicine, Charles and Jane Pak Center for Mineral Metabolism and Clinical Research, University of Texas Southwestern Medical Center, Dallas, United States

**Abstract** Adolescent idiopathic scoliosis (AIS) is the most common spine disorder affecting children worldwide, yet little is known about the pathogenesis of this disorder. Here, we demonstrate that genetic regulation of structural components of the axial skeleton, the intervertebral discs, and dense connective tissues (i.e., ligaments and tendons) is essential for the maintenance of spinal alignment. We show that the adhesion G protein-coupled receptor ADGRG6, previously implicated in human AIS association studies, is required in these tissues to maintain typical spine alignment in mice. Furthermore, we show that ADGRG6 regulates biomechanical properties of tendon and stimulates CREB signaling governing gene expression in cartilaginous tissues of the spine. Treatment with a cAMP agonist could mirror aspects of receptor function in culture, thus defining core pathways for regulating these axial cartilaginous and connective tissues. As *ADGRG6* is a key gene involved in human AIS, these findings open up novel therapeutic opportunities for human scoliosis.

**\*For correspondence:**
ryan.gray@austin.utexas.edu

**Competing interests:** The authors declare that no competing interests exist.

## Introduction

The maturation and homeostasis of a healthy, functional spine require the integration of several musculoskeletal tissues, including bone, cartilage, and connective tissues, muscle, and the peripheral nervous system. The spine consists of a series of segmented bony vertebral bodies linked together by fibrocartilaginous joints named the intervertebral discs (IVDs), which aid in lateral and rotational flexibility and cushioning loading of the spinal column. The spine is further supported by the paraspinal ligaments and provides attachment sites for multiple muscle-tendon insertions. Finally, the structural organization of the axial skeleton aids in the protection of the spinal cord and acts as the central axis of the body to support and maintain posture during movement through the environment (*Bagnat and Gray, 2020*; *Bogduk, 2016*).

Scoliosis is a complex rotational atypical configuration of the spine, commonly diagnosed by a lateral curvature of 10° or greater in the coronal plane (*Cheng et al., 2015*). Scoliosis can be caused by congenital patterning defects of the vertebral column or associated with various neuromuscular or

syndromic disorders (*Murphy and Mooney, 2019*). In contrast, idiopathic scoliosis occurs in otherwise healthy children without associated patterning or neuromuscular conditions. Adolescent idiopathic scoliosis (AIS) is the most common pediatric spine disorder, affecting ~3% of children worldwide (*Cheng et al., 2015*). Clinical interventions for AIS include surgical correction, which aims to halt the progression of severe curves, but which places a high socioeconomic burden on patients and families (*Negrini et al., 2018*). Despite substantial efforts to decipher the pathogenesis of AIS, the molecular genetics and pathology underlying this condition remain ill-defined (*Cheng et al., 2015*; *Newton Ede and Jones, 2016*).

The appearance of spine curvatures in AIS patients without obvious morphological defects of the vertebral column suggests that more subtle defects in one or more regulators of spine stability may contribute to the pathogenesis of this disorder. Indeed, subtle asymmetric morphological changes of the vertebral bodies at the apical region of the curve have been observed in some AIS patients, which may initiate spine curvature (*Lam et al., 2011*; *Liljenqvist et al., 2002*). In addition, changes in cell density and glycosaminoglycan composition (*Shu and Melrose, 2018*; *Urban et al., 2001*), altered ultrastructure of collagen and elastic fibers (*Akhtar et al., 2005*), and increased incidence of endplate-oriented disc herniations (i.e., Schmorl's nodes) (*Buttermann and Mullin, 2008*) may also contribute to the onset of scoliosis in AIS patients.

Genome-wide association studies and subsequent meta-analysis of multiethnic AIS cohorts strongly implicate the *ADGRG6* gene for susceptibility to scoliosis (*Kou et al., 2013*; *Kou et al., 2018*). ADGRG6 (also called GPR126) is a member of the adhesion G protein-coupled receptor family of proteins, many of which display canonical intercellular signaling function via activation of cyclic adenosine monophosphate (cAMP) (*Hamann et al., 2015*; *Langenhan et al., 2016*). In addition, ADGRG6 has been shown to function through canonical cAMP signaling to regulate Schwann cell differentiation and myelination and the regulation of inner ear development (*Geng et al., 2013*; *Monk et al., 2009*; *Monk et al., 2011*). *Adgrg6* null mutant mice are embryonic lethal, with a few survivors displaying cardiac defects and severe joint contractures, but die before weaning (*Mogha et al., 2013*; *Monk et al., 2011*; *Waller-Evans et al., 2010*). However, conditional ablation of *Adgrg6* in osteochondral progenitor cells generated a reproducible genetic mouse model that displayed postnatal-onset scoliosis without causing obvious patterning defects of the vertebral column (*Karner et al., 2015*). However, which elements of the spine and signaling pathways are essential for maintaining normal spine alignment and how treatment of these deficiencies may alleviate pathology and distress in patients remains poorly understood.

This work identifies a unique target of ADGRG6 signaling whereby the ligaments and tendons and IVDs act synergistically to maintain spine alignment rather than bony tissues. We define signaling mechanisms downstream of ADGRG6 that advance the understanding of the pathogenesis of AIS using mouse models of AIS with a high degree of construct validity (*Willner, 1984*).

## Results

### *Adgrg6*/ADGRG6 is expressed in the IVD and supraspinous ligaments

Multiethnic genome-wide association studies show that intronic variants in *GPR126/ADGRG6* locus are associated with AIS in humans (*Kou et al., 2013*; *Kou et al., 2018*). Conditional knockout of *Adgrg6* in osteochondral progenitor cells (*Col2a1-Cre; Adgrg6^{f/f}*) models the timing and pathology of AIS in mouse (*Karner et al., 2015*). Given that *Col2a1-Cre; Adgrg6^{f/f}* mutant mice display a similar phenotype and underlying molecular genetic basis associated with AIS in humans, this mouse model is uniquely suited to study the initiation, progression, and pathogenesis of AIS. However, osteochondral progenitor cells give rise to multiple skeletal elements; thus, the identity and pathology of the tissue(s) that precipitate the onset and progression of AIS in *Col2a1-Cre; Adgrg6^{f/f}* mutant mice were unresolved.

To address the pathogenesis of AIS observed in *Col2a1-Cre; Adgrg6^{f/f}* mutant mice, we first assayed the expression pattern of *Adgrg6* in neonatal mouse spine using fluorescent in situ hybridization, which showed expression in the vertebral growth plate, nucleus pulposus, and the cartilaginous endplate (*Figure 1A–C''*). We also observed *Adgrg6* expression in cells located in the trabecular bone region (*Figure 1A*), throughout the annulus fibrosus (*Figure 1C', C''*), and in the connective tissue surrounding the IVD (the outermost annulus fibrosus, white arrows, *Figure 1C'*),

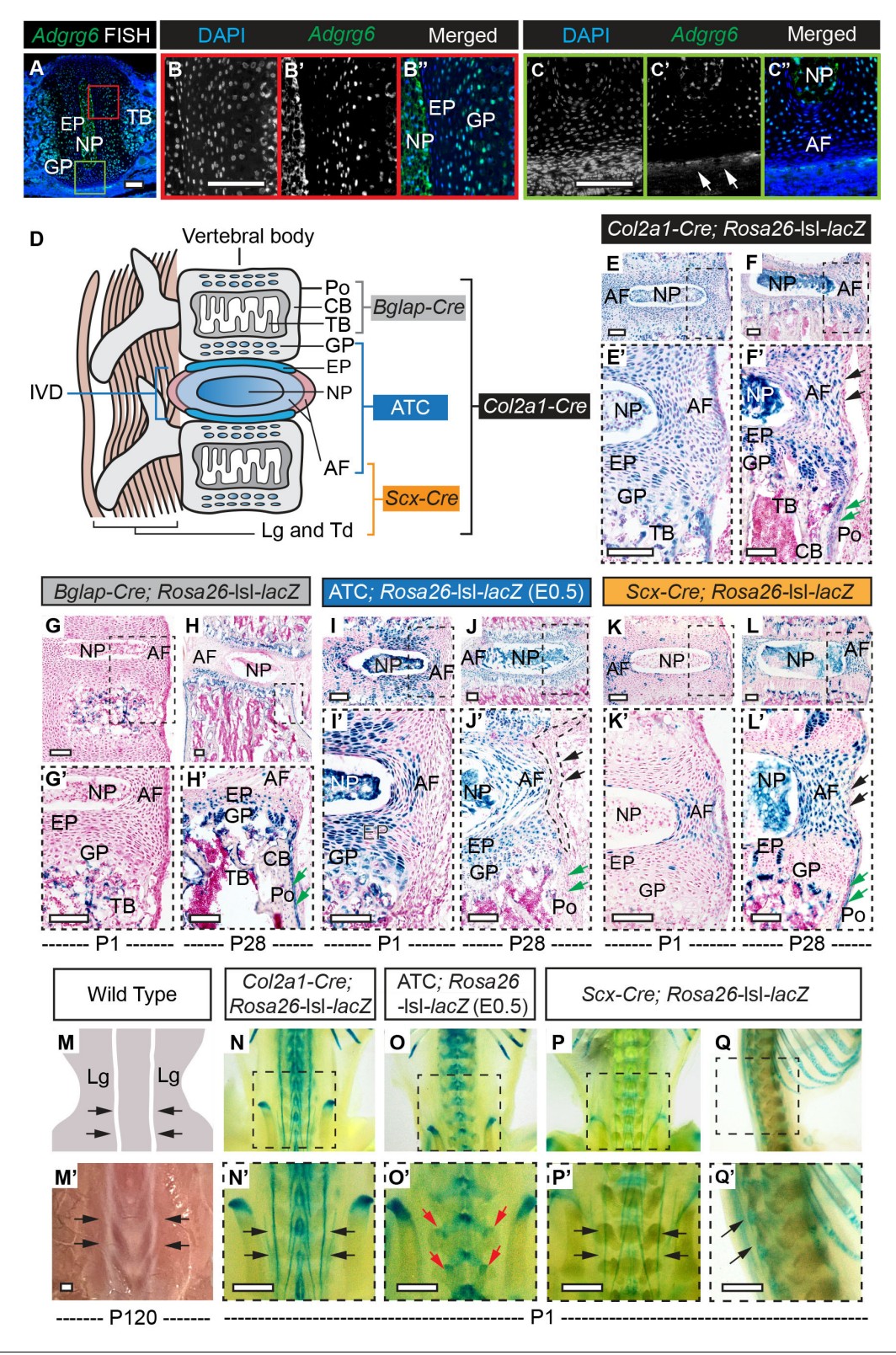

**Figure 1.** β-Galactosidase staining of *Rosa26*-lsl-*lacZ* reporter mice recombined with different Cre strains. (**A–C"**) Fluorescent in situ hybridization (FISH) analysis of *Adgrg6* on thoracic spine sections of wild-type mice at P1. *Adgrg6* signal is detected in the GP, NP, and EP (**B', B"**). *Adgrg6* is also expressed in TB, AF (**A, C', C"**), and the outmost AF (white arrows, **C'**); *n* = 3 mice for each group. (**D**) Schematic of Cre targeting outlined in this study. Generally, *Bglap-Cre* targets bony tissues, ATC targets cartilaginous tissues, *Scx-Cre* targets dense connective tissues, and *Col2a1-Cre* targets all these

*Figure 1 continued on next page*

*Figure 1 continued*

tissues. (**E–F′**) β-galactosidase staining of *Col2a1-Cre; Rosa26-lsl-lacZ* spine sections at P1 (**E, E′**) and P28 (**F, F′**). *Col2a1-Cre* targets NP, AF, EP, and GP, as well as some cells in the bony tissues at both P1 and P28. Note that *Col2a1-Cre* also targets the outmost AF (black arrows, **F′**) and the periosteum (Po) (green arrows, **F′**) at P28. (**G–H′**) β-Galactosidase staining of *Bglap-Cre; Rosa26-lsl-lacZ* spine sections at P1 (**G, G′**) and P28 (**H, H′**). *Bglap-Cre* targets TB at P1 (**G, G′**), but targets TB (**H′**), CB (**H′**), and Po (green arrows, **H′**) at P28. *Bglap-Cre* also targets hypertrophic cells within the EP and GP (**H′**). (**I–J′**) β-Galactosidase staining of ATC; *Rosa26-lsl-lacZ* (Dox-induced from E0.5–P20) spine sections at P1 (**I, I′**) and P28 (**J, J′**). ATC targets most cells in NP, AF, EP, and GP at both time points. Note that ATC did not target the outmost AF (black dash line and arrows, **J′**), nor the Po (green arrows, **J′**). (**K–L′**) β-Galactosidase staining of *Scx-Cre; Rosa26-lsl-lacZ* spine sections at P1 (**K, K′**) and P28 (**L, L′**). *Scx-Cre* targets AF at P1 (**K′**) and P28 (**L′**), and also recombines in several cells of NP, EP, and GP (**L′**) at P28. Note that *Scx-Cre* also targets some cells of the outmost AF (black arrows, **L′**) and the Po (green arrows, **L′**) at P28. An illustration (**M**) and a bright-field image (**M′**) of the dorsal side of a wild-type mouse show the supraspinous ligaments (black arrows, **M, M′**). (**N–Q′**) Whole-mount β-galactosidase staining of Rosa26-lsl-lacZ reporter mice at P1. Dorsal view of *Col2a1-Cre; Rosa26-lsl-lacZ* mouse is shown in (**N, N′**). Supraspinous ligaments are indicated with black arrows (**N′**). Dorsal view of ATC; *Rosa26-lsl-lacZ* mouse (Dox-induced from E0.5) is shown in (**O, O′**). Facet joints are indicated with red arrows (**O′**). Dorsal and sagittal views of the whole-mount *Scx-Cre; Rosa26-lsl-lacZ* mice are shown in (**P, P′**) and (**Q, Q′**), respectively. Supraspinous ligaments are indicated with black arrows (**P′, Q′**). n = 3 mice in each group. Scale bars: 100 μm in (**A–C**) and (**E–L′**); 1 mm in (**M′–Q′**). AF: annulus fibrosis; EP: endplate; GP: growth plate; NP: nucleus pulposus; TB: trabecular bone; CB: cortical bone; Po: periosteum; Lg: ligament; Td: tendon.

The online version of this article includes the following figure supplement(s) for figure 1:

**Figure supplement 1.** Immunohistochemistry (IHC) analyses of ADGRG6 and β-galactosidase staining of *Rosa26-lsl-lacZ* reporter mice recombined with different Cre strains.

which are recapitulated by immunohistochemistry (IHC) using an ADGRG6 antibody in neonatal and perinatal mouse spines (***Figure 1—figure supplement 1A–B′′***).

## Conditional Cre recombination of the spine in the mouse: osteochondral progenitor cells

The *Col2a1-Cre* strain we used recombines in osteochondral progenitor cells (***Long et al., 2001***), which give rise to multiple lineages vital for building the spine and involved in the regulation of spine alignment (***Liu et al., 2019***; ***Zheng et al., 2019***; ***Figure 1D***). To map the recombination of the *Col2a1-Cre* strain in the spine, we crossed it with a *Rosa26-lsl-lacZ* reporter (R26R) strain (***Soriano, 1999***). *Col2a1-Cre; Rosa26-lsl-lacZ* mice showed strong recombination in all the bone- and cartilage-forming lineages of the spine, including the cartilaginous endplate, nucleus pulposus, the entire annulus fibrosus, and the vertebral growth plate (***Figure 1E–F′***). The *Col2a1-Cre* strain also targeted the periosteum, some bone-forming cells in the trabecular and cortical bone (***Figure 1E–F′***), as well as the ribs and sternum (***Figure 1—figure supplement 1L, O***). Whole-mount β-galactosidase staining of a *Col2a1-Cre; Rosa26-lsl-lacZ* mouse at P1 showed that *Col2a1-Cre* also recombined in some dense connective tissues (i.e., ligaments and tendons) along the spine (***Figure 1N, N′***), such as the supraspinous ligaments flanking the spine (black arrows, ***Figure 1M′, N′***). Coronal and transverse sections of the *Col2a1-Cre; Rosa26-lsl-lacZ* mouse spine showed that most cells in the supraspinous ligaments were LacZ positive (***Figure 1—figure supplement 1G–K***), and IHC analyses on adjacent sections showed that these cells also expressed ADGRG6 (***Figure 1—figure supplement 1H′, I′, and K′***). To understand the knockdown efficiency of *Adgrg6* in *Col2a1-Cre; Adgrg6^{f/f}* mice, we performed IHC analyses and observed significantly reduced expression of ADGRG6 in endplate and growth plate in the mutant mice at P20 (***Figure 1—figure supplement 1D–F***). We also observed reduced expression in the annulus fibrosis of mutant mice (p=0.068, ***Figure 1—figure supplement 1F***), which was especially apparent in scoliotic individuals (p=0.0383, red dots; ***Figure 1—figure supplement 1F***). Taken together, these data demonstrate *Col2a1-Cre; Adgrg6^{f/f}* effectively ablates *Adgrg6*/ADGRG6 expression throughout the spinal column.

## Conditional Cre recombination of distinct axial skeletal elements: bone, cartilaginous, and dense connective tissues

We next sought to genetically dissect which of these tissues require ADGRG6 function for maintaining spine alignment in the mouse. For this, we obtained several Cre driver mouse strains reported to recombine in distinct tissue elements of the spine (***Figure 1D***). To assay ADGRG6 function in the bone, we used *Bone gamma-carboxyglutamic acid-containing protein (Bglap)-Cre* strain to target recombination in mature osteoblasts (***Zhang et al., 2002***). β-Galactosidase staining of spine sections

of *Bglap-Cre; Rosa26*-lsl-*lacZ* mice at P1 showed that *Bglap-Cre* specifically targeted mature osteoblasts in the newly forming trabecular bone (*Figure 1G, G'*). At P28, LacZ expression was observed in trabecular bone, cortical bone, periosteum, and some hypertrophic cells in the vertebral growth plate and cartilaginous endplate (*Figure 1H, H'*).

To assay ADGRG6 function specifically in committed chondrocyte lineages, we used an *Aggrecan* enhancer-driven, *Tetracycline-inducible Cre* strain (ATC) (*Dy et al., 2012*; *Liu et al., 2019*). β-Galactosidase staining of skeletal sections from ATC; *Rosa26*-lsl-*lacZ* mice induced at embryonic (E)0.5 showed robust recombination in cartilaginous tissues of the IVD, including within the cartilaginous endplate, nucleus pulposus, vertebral growth plate, and most cells of the inner annulus fibrosus (*Figure 1I–J'*). Interestingly, the outermost annulus fibrosus and the periosteum were not effectively recombined by ATC (*Figure 1J'*, black arrows and green arrows, respectively). A similar recombination pattern was observed in ATC; *Rosa26*-lsl-*lacZ* mice induced at P1 (*Liu et al., 2019*). Whole-mount β-galactosidase staining of ATC; *Rosa26*-lsl-*lacZ* mice at P1 showed recombination of ATC in the facet joints of the spine (red arrows, *Figure 1O'*) and the distal cartilaginous portion of the ribs (red arrow, *Figure 1—figure supplement 1M*). However, it did not target the bony portion of the ribs and sternum (yellow arrows, *Figure 1—figure supplement 1M, P*). In summary, both embryonic and perinatal induction of ATC-driven recombination effectively targets committed cartilaginous tissues without significant recombination in bone and connective tissues of the spine.

The supraspinous ligaments of the spine are recombined by the *Col2a1-Cre* strain (*Figure 1N, N'*) but not by the ATC strain (*Figure 1O, O'*). Therefore, we hypothesized that these dense connective tissues might play a synergistic role with the IVD to maintain spine alignment. In addition, the outermost annulus fibrosus (black arrows, *Figure 1F'*), as well as some cells in the periosteum (green arrows, *Figure 1F'*), which collectively express ADGRG6 (*Figure 1—figure supplement 1A–B", H', I', and K'*), yet are not recombined by the ATC strain (*Figure 1J'*). To specifically target dense connective tissues of the spine, we utilized the *Scx-Cre* strain, which recombines in tendon and ligament progenitor cells (*Blitz et al., 2009*). Whole-mount β-galactosidase staining of *Scx-Cre; Rosa26*-lsl-*lacZ* mice at P1 revealed that *Scx-Cre* targeted the same supraspinous ligaments (black arrows, *Figure 1P, P'*), as is observed in the *Col2a1-Cre; Rosa26*-lsl-*lacZ* mice (black arrows, *Figure 1N, N'*). In addition, β-galactosidase staining of spine sections from *Scx-Cre; Rosa26*-lsl-*lacZ* mice revealed sparse targeting of the annulus fibrosus at P1 (*Figure 1K, K'*) but robust targeting of the annulus fibrosus at P28 (*Figure 1L–L'*). *Scx-Cre* also showed sparse recombination within a subset of cells in the nucleus pulposus, endplate, growth plate (*Figure 1L–L'*), and the periosteum (green arrows, *Figure 1L'*) at P28. Notably, *Scx-Cre* did not target the bony portion of the ribs or sternum (yellow arrows, *Figure 1—figure supplement 1N, Q*) but recombined in some cells of the cartilaginous portion of the ribs (red arrow, *Figure 1—figure supplement 1N*). In summary, we show that *Scx-Cre* recombines in dense connective tissues of the spine, with only minor contributions to other structural elements of the spine.

## Ablation of *Adgrg6* in osteochondral progenitor cells models progressive AIS in mouse

To determine the natural history of scoliosis in *Col2a1-Cre; Adgrg6$^{f/f}$* mice, we performed longitudinal X-ray analysis at postnatal day 10 (P10) out to P120. Both wild-type (WT) Cre (-) and heterozygous Cre (+) (*Col2a1-Cre; Adgrg6$^{f/+}$*) control littermates showed typical patterning of a straight spinal column at all time points (*Figure 2A–D'* and *Figure 2—figure supplement 1A, A'*). *Col2a1-Cre; Adgrg6$^{f/f}$* mice were indistinguishable from littermate controls at birth and displayed typical patterning and alignment of the spine at P10 (n = 23) (*Figure 2E, E'*). However, at P20, we first observed postnatal-onset spine curvatures (*Figure 2F, F'*), which could be progressive in some *Col2a1-Cre; Adgrg6$^{f/f}$* animals (*Figure 2F–H'*). Approximately 60% of *Col2a1-Cre; Adgrg6$^{f/f}$* mice exhibited scoliosis at P20 (52.1%; n = 23), P40 (60.8%; n = 23), and P120 (58.8%; n = 17) (*Figure 2I, K, L*), with the apex of the curvature centered at thoracic vertebrae (T)8 and T9 (*Figure 2J*). Scoliosis is defined by a Cobb angle measure greater than 10° in the coronal plane (*Cobb, 1958*). The mean Cobb angle increased in *Col2a1-Cre; Adgrg6$^{f/f}$* mice compared with control littermates (*Figure 2I*), ranging from 11° to 46° in scoliotic individuals (*Figure 2I, L*). In comparison, only one Cre (+) control mouse (n = 18) had a mild curve of 13.8° at P120, with the remainder displaying no alteration in spine alignment (*Figure 2I, K* and *Figure 2—figure supplement 1B*). We observed scoliosis in both 37.5% (n = 8) of male and 66.6% (n = 15) of female mutant mice, with

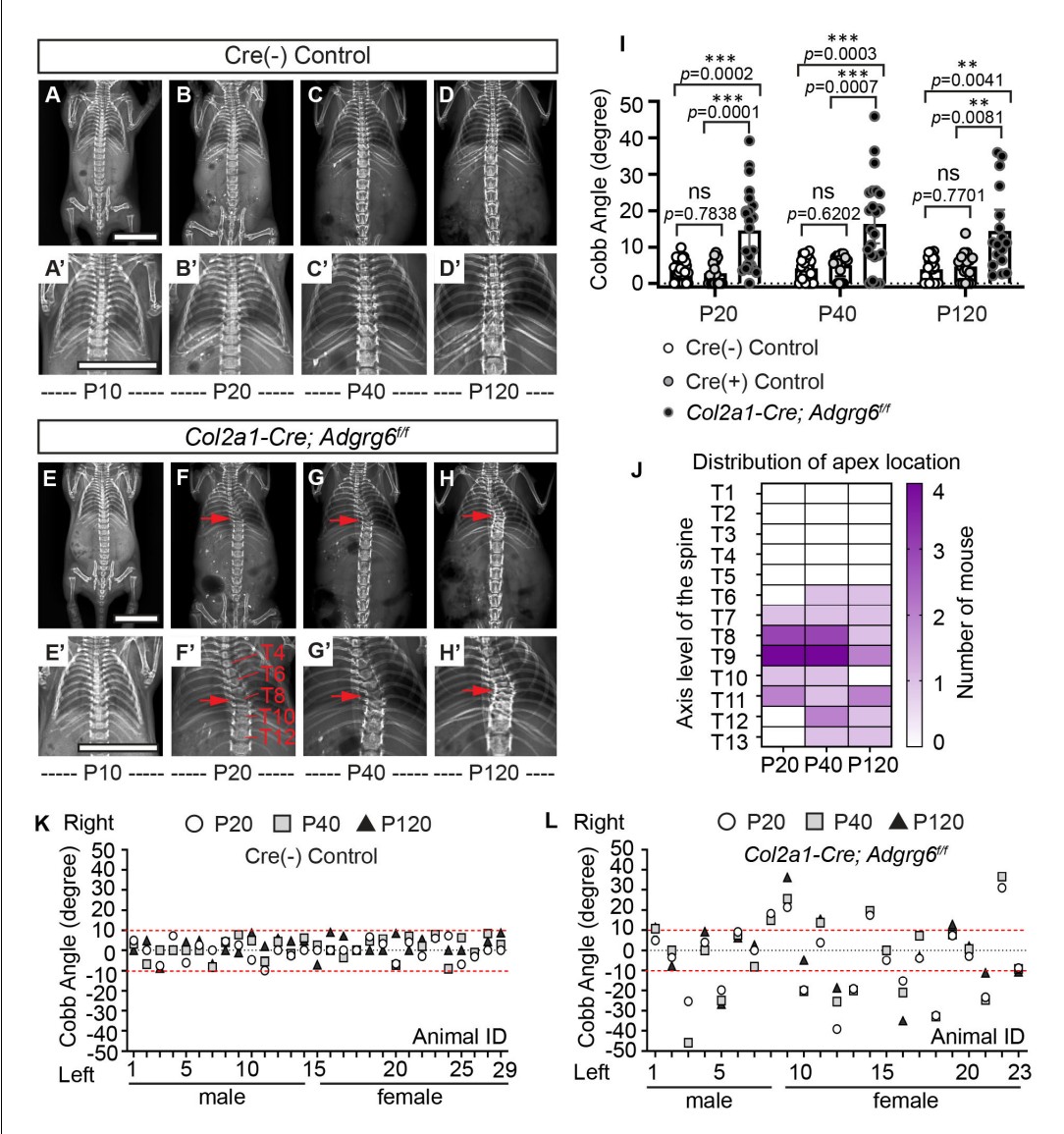

**Figure 2.** Ablation of *Adgrg6* in osteochondral progenitor cells models progressive adolescent idiopathic scoliosis (AIS) in mouse. (**A–D'**) Longitudinal X-ray analysis of a representative Cre (-) control mouse at P10 (**A, A'**), P20 (**B, B'**), P40 (**C, C'**), and P120 (**D, D'**). (**E–H'**) Longitudinal X-ray analysis of a representative *Col2a1-Cre; Adgrg6^{f/f}* mutant mouse at P10 (**E, E'**), P20 (**F, F'**), P40 (**G, G'**), and P120 (**H, H'**), showing adolescent-onset (**F**) and progressive (**F–H**) thoracic scoliosis, with the apex of scoliosis indicated (red arrows; **F–H'**). Thoracic (T) vertebrae are labeled in (**F'**). (**I**) Longitudinal Cobb angle analysis; Cre (-) control mice, *n* = 29 mice; Cre (+) control mice, *n* = 18, 17, and 18 mice at P20, P40, and P120, respectively; and *Col2a1-Cre; Adgrg6^{f/f}* mutant mice, *n* = 23, 23, and 17 mice at P20, P40, and P120, respectively. Dots are plotted with mean ± 95% CI. The statistical difference is evaluated by two-way ANOVA followed by Tukey's multiple comparison test. The p-value for each comparison is shown. (**J**) Heat map of the apex distribution of the scoliotic *Col2a1-Cre; Adgrg6^{f/f}* mice at P20, P40, and P120. The heat map is plotted with the axis level of the thoracic (T) spine (T1–T13, left axis) and the number of mice with scoliosis (right axis) with apex observed at each level. The apex of scoliosis is distributed along the middle to lower thoracic spine (T6–T13), with hotspots at T8 and T9. (**K, L**) Cobb angle values for all the Cre (-) control mice (**K**) and *Col2a1-Cre; Adgrg6^{f/f}* mice (**L**) showed in (**I**). Thresholds of scoliosis (Cobb angle >10°) are indicated with two red dot lines. Scale bars: 10 mm.

The online version of this article includes the following source data and figure supplement(s) for figure 2:

**Source data 1.** Cobb angle measurements of mice with *Adgrg6* ablation in osteochondral progenitor cells.

**Figure supplement 1.** X-ray analysis and Cobb angle measurements of Cre (+) control mice.

both right- or left-ward curvatures (*Figure 2L*). We did not observe an obvious sex bias for the incidence of scoliosis (Fisher exact test, p=0.2213).

## Wedging of the IVD is concurrent with the onset of scoliosis in *Col2a1-Cre; Adgrg6^{f/f}* mutant mice

Wedging of the vertebrae and the IVD within regions of acute spine curvature is observed in human AIS (*Little et al., 2016*; *Newton Ede and Jones, 2016*). To determine if similar morphological changes of the spine were associated with the initiation and progression of scoliosis, we harvested the thoracic spine (T5–T12) of *Col2a1-Cre; Adgrg6^{f/f}* and control littermates at P20 and P120. During the initiation of scoliosis (P20), we observed wedging of IVDs close to the apex of the curvature in *Col2a1-Cre; Adgrg6^{f/f}* mice with scoliosis, associated with a shift of the nucleus pulposus towards the convexity of the curve (*Figure 3B*). The large vacuolated cells (magenta stained/notochordal cells) shifted towards the convex side, while the smaller non-vacuolated cells (blue-stained/chondrocyte-like cells) distributed in the middle and concave side (*Figure 3B, B''*), in contrast to the uniform distribution of these cells with the IVD typical of control littermates (*Figure 3A, A''*). We observed more vertical alignment of the annulus fibrosis lamellae in *Col2a1-Cre; Adgrg6^{f/f}* mice along the convex side (white dash lines, *Figure 3A', B'*), with flatter lamellae and an elongated inner annulus fibrosus at the concave side the IVD (white dash lines, *Figure 3A, B*).

Older *Col2a1-Cre; Adgrg6^{f/f}* scoliotic mice (P120) showed a more pronounced shift of the nucleus pulposus to the convex side (*Figure 3—figure supplement 1D, D''*), with vertical alignment of the annulus fibrosus lamellae (*Figure 3—figure supplement 1D, D'*) and the endplate and growth plate tissues appeared thinner (*Figure 3—figure supplement 1C''', D'''*). Analysis of the cartilaginous endplate and the vertebral growth plate in *Col2a1-Cre; Adgrg6^{f/f}* mice showed no obvious histopathology at P20 (*Figure 3A, A''', B, B''*). Despite wedging of IVDs close to the apex of the curvature in *Col2a1-Cre; Adgrg6^{f/f}* mice, no apparent defects in the size or patterning of the vertebrae, growth plates, or IVD tissues were observed during the initiation of scoliosis at P20 (*Figure 3—figure supplement 1A, B*), nor prior to the onset of scoliosis at P10 (*Figure 3—figure supplement 1E–F''*). However, we occasionally observed bulging of the cortical bone outward on the concave side of the curve at P20 (50%; *n* = 4; black arrow, *Figure 3—figure supplement 1B*). Interestingly, the IVDs lying proximal or distal to prominent spine curvatures in *Col2a1-Cre; Adgrg6^{f/f}* mice exhibited typical nucleus pulposus and annulus fibrosus tissues compared with littermate control tissues (*Figure 3—figure supplement 1A', B'*), suggesting that local unbalance of forces at the thoracic spine drives the alterations in tissue architecture.

## *Col2a1-Cre; Adgrg6^{f/f}* mutant mice show a reduction in SOX9 protein expression without significant alterations of IVD matrix composition at the onset of scoliosis

Analysis of the IVD and growth plate in human AIS patients has demonstrated cellularity alterations, changes in the typical expression of anabolic and catabolic genes, and reduced proteoglycan content (*Newton Ede and Jones, 2016*; *Roberts et al., 1993*). Analysis of the transcriptional and epigenetic changes in *Col2a1-Cre; Adgrg6^{f/f}* mutant IVD tissue demonstrated altered regulation of several anabolic factors and extracellular matrix (ECM) components expressed in healthy cartilaginous tissues, including *Sox5*, *Sox6*, and *Sox9* genes (*Liu et al., 2019*; *Makki et al., 2021*). We assayed SOX9, an essential transcriptional regulator of cartilage and IVD homeostasis in mouse (*Henry et al., 2012*), which showed strong expression throughout the IVD in littermate control mouse spine at P20 (*Figure 3C–C''*) and P10 (*Figure 3—figure supplement 1I*). In contrast, *Col2a1-Cre; Adgrg6^{f/f}* mice showed reduced SOX9 expression throughout the IVD at P20 in both scoliotic and non-scoliotic mutant mice (*Figure 3D–D'', G*). Interestingly, SOX9 expression is not reduced in the IVD of P10 mutant mice (*Figure 3—figure supplement 1J, L*), suggesting that loss of SOX9 expression in the IVD is associated with the initiation of scoliosis.

SOX9 directly regulates type II collagen (COLII) expression (*Bell et al., 1997*). Analysis of COLII expression in the IVD shows a concentration gradient of high expression in the nucleus pulposus, decreasing the annulus fibrosus outwards, while type I collagen (COLI) is primarily expressed in the annulus fibrosus (*Beard et al., 1980*; *Mader et al., 2016*). We observed regular expression of COLII in the IVDs at P20 regardless of genotype (*Figure 3E, F*); however, the typical distribution of COLI-

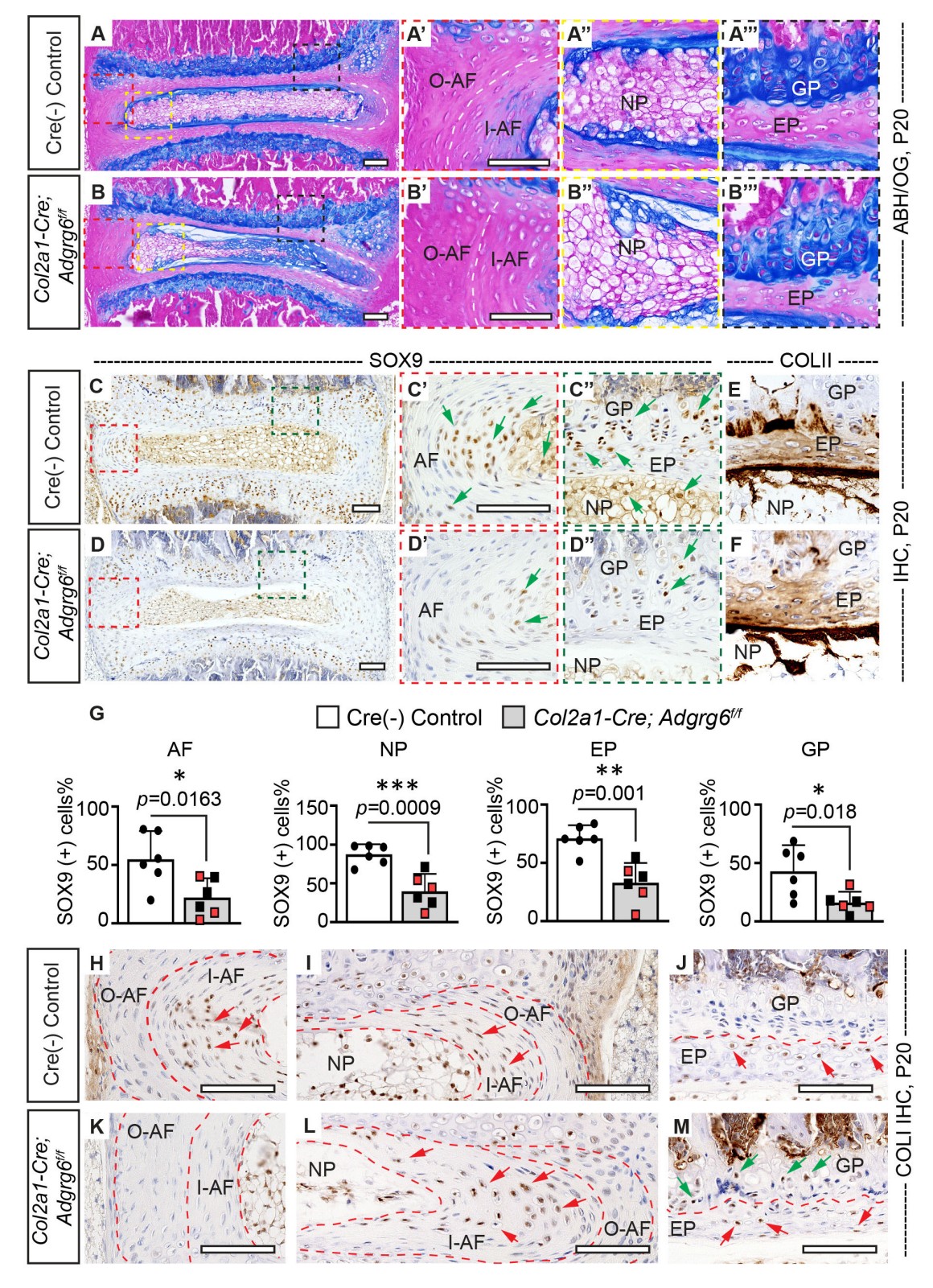

**Figure 3.** *Col2a1-Cre; Adgrg6^{f/f}* mice display alterations in intervertebral discs (IVD). (A–B''') Midline sectioned IVD sections from Cre (-) control (A–A''') and scoliotic *Col2a1-Cre; Adgrg6^{f/f}* mutant mice (B–B''') stained with Alcian Blue Hematoxylin/Orange G (ABH/OG) at P20. The IVD close to the apex of the curve in the *Col2a1-Cre; Adgrg6^{f/f}* mice is mildly wedged, associated with a shift in the position of the nucleus pulposus towards the convex side of the curve (B, B''). The inner AF (I–AF) and outer AF (O–AF) in the convex side of the *Col2a1-Cre; Adgrg6^{f/f}* mice are composed of more vertical lamellae

*Figure 3 continued on next page*

*Figure 3 continued*

compared with that in Cre (-) control mice (white dash lines, **A'**, **B'**). The AF in the concave side of the *Col2a1-Cre; Adgrg6^{f/f}* mice is elongated and composed with flatter lamellae, compared with that in Cre (-) control mice (white dash lines, **A**, **B**). No overt structural defects are observed in the EP and GP of the *Col2a1-Cre; Adgrg6^{f/f}* mice (**B'''**). Cre (-) control mice, *n* = 4 mice; scoliotic *Col2a1-Cre; Adgrg6^{f/f}* mutant mice, *n* = 3 mice. (**C–G**) Immunohistochemistry (IHC) analyses of common anabolic markers of healthy IVD at P20. *Col2a1-Cre; Adgrg6^{f/f}* mutant mice display reduced expression of SOX9 in AF, EP, and GP (green arrows, **C'–D''**), which is quantified in (**G**). *Col2a1-Cre; Adgrg6^{f/f}* mice show a normal expression pattern of COLII compared with the controls (**E**, **F**). Cre (-) control mice, *n* = 6 mice; *Col2a1-Cre; Adgrg6^{f/f}* mutant mice, *n* = 6 mice (three scoliotic mice, three non-scoliotic mice). Bars are plotted with mean and 95% CI. Each dot represents one mouse analyzed. Scoliotic *Col2a1-Cre; Adgrg6^{f/f}* mutant mice are marked in red in (**G**). The statistical difference is evaluated by a two-tailed Student's *t*-test. The p-value for each comparison is shown. (**H–M**) IHC analyses of COLI at P20. COLI is mainly expressed in the I-AF (**H**, **I**) and some cells of the EP (**J**) in the control mice (indicated with red arrows). No obvious expression of COLI is observed in the O-AF or I-AF in the convex side of the mutant IVD (**K**), but robust expression is observed in the elongate I-AF in the concave side of the mutant IVD (red arrows, **L**). Notably, some cells in the GP of the *Col2a1-Cre; Adgrg6^{f/f}* mice also express COLI (green arrows, **M**). Cre (-) control mice, *n* = 4 mice; scoliotic *Col2a1-Cre; Adgrg6^{f/f}* mutant mice, *n* = 3 mice. Scale bars: 100 μm. AF: annulus fibrosis; I-AF: inner annulus fibrosis; O-AF: outer annulus fibrosis; EP: endplate; GP: growth plate; NP: nucleus pulposus.

The online version of this article includes the following source data and figure supplement(s) for figure 3:

**Source data 1.** Quantification of SOX9 positive cells.
**Figure supplement 1.** Loss of *Adgrg6* in osteochondral progenitor cells leads to alternations in spinal elements.

positive (COLI+) cells was altered *Col2a1-Cre; Adgrg6^{f/f}* mice with scoliosis (*Figure 3H–M*). In control IVDs, COLI+ cells are observed in the inner annulus fibrosus and sparsely in the cartilaginous endplate (red arrows, *Figure 3H–J*). However, *Col2a1-Cre; Adgrg6^{f/f}* mutants showed a reduction in COLI+ cells in the inner annulus fibrosus at the convex side of the curvature (*Figure 3K*), concomitant with increased COLI+ cells in the inner annulus fibrosus on the concave side (red arrows, *Figure 3L*). Interestingly, the growth plate consistently showed increased COLI+ cells in *Col2a1-Cre; Adgrg6^{f/f}* mice (green arrows, *Figure 3M*).

Proteoglycans are a broad category of macromolecules characterized by a core protein modified by a diverse array of complex glycosaminoglycan (GAG) chains (*Iozzo and Schaefer, 2015*) bulk of ECM of the IVD. We utilized Strong-Anion-Exchange high-pressure liquid chromatography (SAX-HPLC) to quantify proteoglycan species within the IVDs isolated from Cre (-) control and *Col2a1-Cre; Adgrg6^{f/f}* mice at P20. The main GAG species detected are various chondroitin sulfate (CS) species and a minor amount of hyaluronan (HA) species. However, we observed no significant changes in GAG quantity or alterations in specific GAG species (*Supplementary file 1*), demonstrating that loss of *Adgrg6* did not alter the expression of a significant class of proteoglycan species in the IVD at the initiation of scoliosis.

In summary, these results show that loss of *Adgrg6* in osteochondral progenitor cells leads to alterations in the typical expression pattern of SOX9+ and COLI+ cells within the IVDs. In contrast, alterations of bulk ECM components, including COLII and GAG proteoglycans, were not significantly affected at the onset of scoliosis, suggesting that even minor alterations in SOX9 expression in the postnatal IVD of mice may be sufficient to initiate biomechanical instability leading to disc wedging and driving the onset of spine curvature, without grossly affecting tissue architecture and composition of the major ECM components of the IVD.

## ADGRG6 regulates cAMP/CREB signaling-dependent gene expression in the IVD

Next, we set out to understand how ADGRG6 regulates the homeostasis of the IVD. Given the established role of ADGRG6 for control of cAMP signaling in other contexts (*Geng et al., 2013*; *Monk et al., 2009*; *Monk et al., 2011*), we first analyzed the expression pattern of Serine-133 phosphorylated CREB (pCREB)—the active form of cAMP response element-binding protein (CREB)—in spine tissues. pCREB expression is typically enriched throughout cartilaginous elements of the spine, including the cartilaginous endplate and nucleus pulposus (*Figure 4A–A''*, *C*, *Figure 3—figure supplement 1G, G'*). It is also expressed in some cells of the periosteum (*Figure 4A''', C*). In contrast, *Col2a1-Cre; Adgrg6^{f/f}* mice showed a decrease in pCREB+ cells in these tissues at P20 (*Figure 4B–B''', C*), indicating that *Adgrg6* is essential for regulating pCREB expression in cartilaginous tissues of the spine. Furthermore, IHC showed reduced pCREB expression in the nucleus pulposus in *Col2a1-Cre; Adgrg6^{f/f}* mutants at P10 (*Figure 3—figure supplement 1H, H', K*), suggesting that

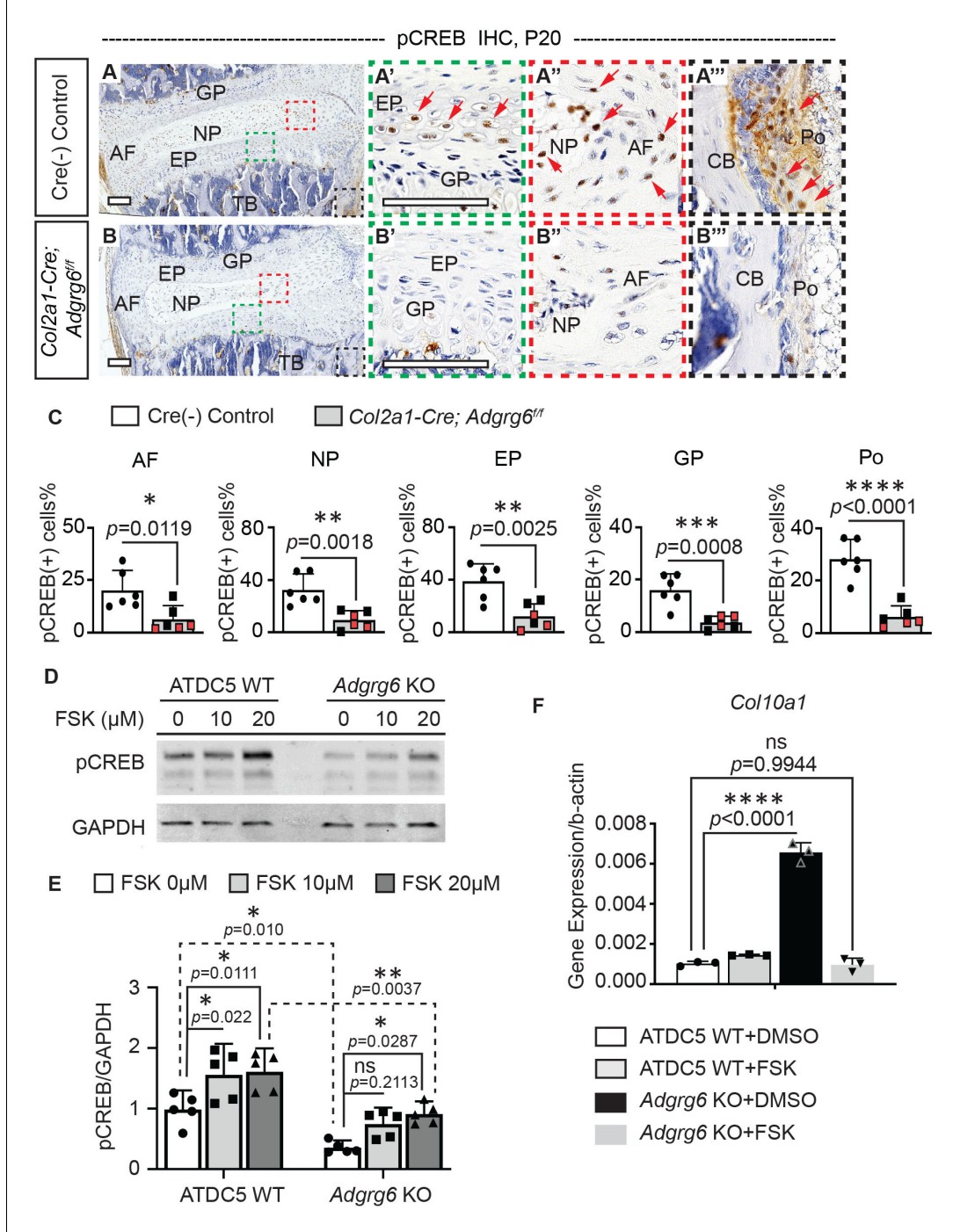

**Figure 4.** ADGRG6 regulates cAMP/CREB signaling-dependent gene expression in cartilaginous lineages. (**A–C**) Immunohistochemistry (IHC) analyses of pCREB in mouse spine sections at P20. Cre (-) control mice exhibit pCREB-positive cells in AF, NP, EP, GP, and Po (red arrows, **A–A'''**). *Col2a1-Cre; Adgrg6^{f/f}* mutant mice show reduced expression of pCREB in all these tissues (**B–B'''**). pCREB-positive cells are quantified in (**C**). Cre (-) control mice, *n* = 6 mice; *Col2a1-Cre; Adgrg6^{f/f}* mutant mice, *n* = 6 mice (three scoliotic mice, three non-scoliotic mice). Bars are plotted with mean and 95% CI. Each dot represents one mouse analyzed. Scoliotic *Col2a1-Cre; Adgrg6^{f/f}* mutant mice are marked in red in (**C**). The statistical difference is evaluated by a two-tailed Student's *t*-test. The p-values for each comparison are shown. (**D, E**) Representative western blot image (**D**) and densitometry of the western blot images (**E**) on pCREB in both wild-type (WT) and *Adgrg6* KO ATDC5 cell lysates. The expression level of pCREB is significantly decreased in *Adgrg6* KO cells compared with ATDC5 WT cells. Treatment with 20 μM of Forskolin (FSK) for 30 min can stimulate pCREB expression in both ATDC5 WT cells and *Adgrg6* KO cells, though the induction level in *Adgrg6* KO cells is significantly lower than that in ATDC5 WT cells (**E**). *n* = 5 biological replicates. Bars are plotted with mean and 95% CI. The statistical difference is evaluated by two-way ANOVA followed by Tukey's multiple comparison test. The p-value for each comparison is shown. ns: not significant. (**F**) Real-time RT-PCR analyses of *Col10a1* in ATDC5 WT cells and *Adgrg6* KO cells,

*Figure 4 continued on next page*

*Figure 4 continued*

cultured with FSK (2 µM) or DMSO control for 7 days with maturation medium. The increased expression of *Col10a1* in *Adgrg6* KO cells was rescued by FSK treatment. *n* = 3 biological replicates, and the representative result is shown. Bars were plotted with mean and 95% CI. The statistical difference is evaluated by one-way ANOVA followed by Tukey's multiple comparison test. The p-value for each comparison is shown. ns: not significant. Scale bars: 100 µm. AF: annulus fibrosis; EP: endplate; GP: growth plate; NP: nucleus pulposus; TB: trabecular bone; CB: cortical bone; Po: periosteum.

The online version of this article includes the following source data and figure supplement(s) for figure 4:

**Source data 1.** Quantification of pCREB expression.
**Figure supplement 1.** Forskolin treatment can partially rescue the dysregulation of *Sox9* expression but not *Col2a1* and *Acan* expression in *Adgrg6* KO cells.

alteration of CREB signaling emerges before the initiation of scoliosis. We previously demonstrated that *Adgrg6* knockout (KO) ATDC5 chondrogenic cells displayed global disruptions of typical chondrogenic gene expression (*Liu et al., 2019*). In agreement with our findings in the IVD, we show reduced pCREB in *Adgrg6* KO ATDC5 cells compared with unedited control cells (*n* = 5 biological repeats) (*Figure 4D, E*). Given that ADGRG6 can signal through Gs/cAMP in other contexts (*Geng et al., 2013*; *Mogha et al., 2013*; *Monk et al., 2009*), we next asked whether treatment with Forskolin (FSK), a small molecule activator of the adenylyl cyclase, could restore pCREB expression independent of ADGRG6 function in ATDC5 cells. Western analysis showed that the addition of FSK to ATDC5 cell culture could stimulate pCREB expression in both ATDC5 WT cells and *Adgrg6* KO cells, albeit at a reduced level of expression after FSK treatment in *Adgrg6* KO cells (*Figure 4D, E*).

Loss of *Adgrg6* in cartilaginous tissues of the spine led to increased hypertrophy and expression of type X collagen (encoded by *Col10a1*) in cartilaginous endplate and growth plate (*Liu et al., 2019*). CREB signaling is associated with reduced *Col10a1*expression in primary growth plate chondrocytes (*Li et al., 2004*). Interestingly, asymmetric expression of type X collagen in vertebral growth plates has been observed in AIS patient samples (*Wang et al., 2010*). Next, we set out to determine if FSK treatment could rescue altered gene expression outputs due to loss of ADGRG6 function. We found that treatment of ATDC5 cells with a low concentration of FSK (2 µM) for 7 days restored typical expression of *Col10a1* and partially restored *Sox9* expression in *Adgrg6* KO cells (*Figure 4F* and *Figure 4—figure supplement 1*). However, the expression of other affected genes, such as decreased expression of *Acan* and *Col2a1* in *Adgrg6* KO cells, was not restored by this approach (*Figure 4—figure supplement 1*). Taken together, these results support a model where *Adgrg6* is necessary for cAMP-driven CREB signaling in several cartilaginous tissues for the regulation of a subset of essential genes involved in the homeostasis of the IVD.

## *Adgrg6* is dispensable in osteoblast lineages for the development and alignment of the spine

Abnormal bone quality or osteopenia is frequently associated with AIS (*Cheng et al., 2000*; *Cheng et al., 2001*; *Lam et al., 2011*). To examine whether alterations could further explain scoliosis in *Col2a1-Cre; Adgrg6^{f/f}* mice in bone quality, we generated *Bglap-Cre; Adgrg6^{f/f}* mice. All *Bglap-Cre; Adgrg6^{f/f}* mice we assayed showed straight spine alignment at P40 and P120 (*n* = 8) (*Figure 5C, E*), which was typical of both Cre (-) or Cre (+) littermate controls (*Figure 5A, B, D*). Similar results were observed in *Sp7Cre; Adgrg6^{f/f}* mice in which *Adgrg6* was ablated in committed osteoblast precursors (*Figure 5—figure supplement 1C*, *n* = 9). Micro-computed tomography (microCT) analyses revealed the typical architecture of the vertebral bodies and comparable bone mass in the thoracic spines of both Cre (-) control and *Bglap-Cre; Adgrg6^{f/f}* mice at P120 (*Figure 5F–I*), which was confirmed by the bone volume analysis (BV/TV) on both control and mutant vertebral bodies (*Figure 5J*, *n* = 4 for each group). We did not observe histopathological changes of the vertebral bodies or IVDs of the *Bglap-Cre; Adgrg6^{f/f}* mice at P120 (*Figure 5—figure supplement 1D, E*). Real-time RT-PCR analyses revealed very low expression of *Adgrg6* in the Cre (-) control sample compared with the expression of *Col1a1* (0.2% expression of *Col1a1*), which is the most abundant collagen expressed in bone (*Long, 2012*; *Figure 5K*). However, the expression of *Adgrg6* was significantly reduced (~71%) in *Bglap-Cre; Adgrg6^{f/f}* mice (*Figure 5L*). These results show that *Adgrg6* has no apparent role in mature osteoblasts during postnatal bone mass accrual or homeostasis and that alterations in bone quality are not driving scoliosis in *the Col2a1-Cre; Adgrg6^{f/f}* AIS mouse model.

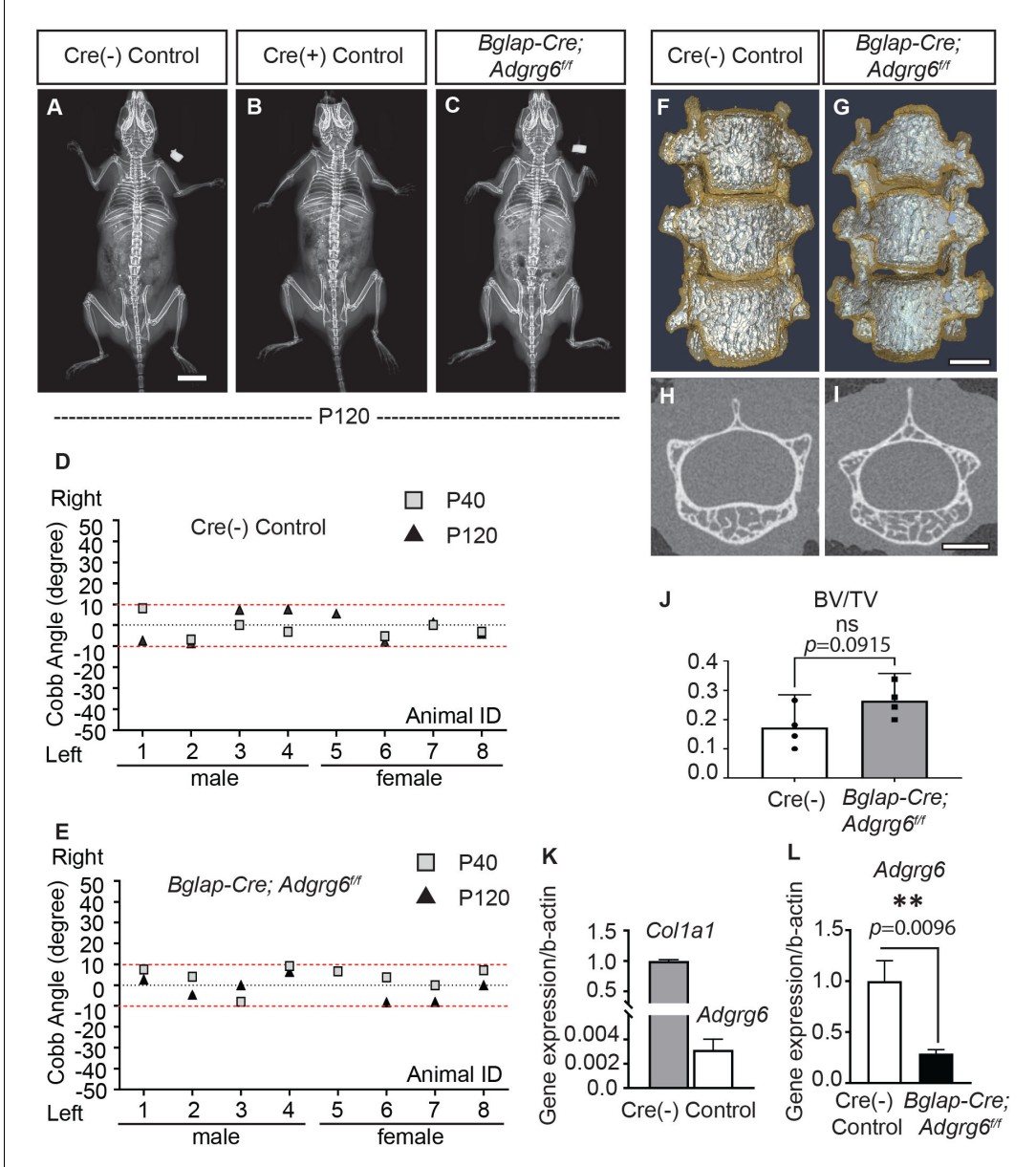

**Figure 5.** Loss of *Adgrg6* in mature osteoblast lineages is dispensable of adolescent idiopathic scoliosis (AIS) development. (A–C) Representative X-ray images of Cre (-) control (A), Cre (+) control (B), and *Bglap-Cre; Adgrg6^{f/f}* mutant (C) mice at P120. (D, E) Longitudinal analyses of Cobb angle values of Cre (-) control mice (D) and *Bglap-Cre; Adgrg6^{f/f}* mice (E) at P40 and P120. Cre (-) control mice, *n* = 8 mice at P40 and P120; *Bglap-Cre; Adgrg6f/f* mutant mice, *n* = 8 and 7 at P40 and P120, respectively. Thresholds of scoliosis (Cobb angle >10°) are indicated with two red dot lines. No *Bglap-Cre; Adgrg6^{f/f}* mice showed scoliosis at P40 (0/8) or P120 (0/7) (E). (F–J) MicroCT scanning of the thoracic region of the spine shows normal morphology of the vertebral bodies in both Cre (-) controls (F) and the *Bglap-Cre; Adgrg6^{f/f}* mice (G). Transverse sections of the microCT three-dimensional reconstruction of the thoracic vertebral body show a comparable bone mass in the control (H) and mutant mice (I). The bone volume per total volume (BV/TV) of the control and mutant mice is shown in (J). *n* = 4 mice for each group. Bars are plotted with mean and 95% CI. The statistical difference is evaluated by a two-tailed Student's *t*-test. The p-value is shown. ns: not significant. (K, L) Real-time RT-PCR analysis of RNA isolated from long bone shows that the expression of *Adgrg6* is very low in bony tissues compared with the expression of *Col1a1* (K). However, the expression of *Adgrg6* was efficiently knockdown in *Bglap-Cre; Adgrg6^{f/f}* mice (L). RNA was isolated and pooled from three mice of each experimental group. Bars are plotted with mean and SD. The statistical difference is evaluated by a two-tailed Student's *t*-test. The p-value for each comparison is shown. Scale bars: 10 mm in (A); 1 mm in (G) and (I).

The online version of this article includes the following source data and figure supplement(s) for figure 5:

**Source data 1.** Characterization of mice with *Adgrg6* ablation in mature osteoblast lineages.

**Figure supplement 1.** Loss of *Adgrg6* in osteoblast lineages leads to no scoliosis or spinal deformity.

## Specific ablation of *Adgrg6* in cartilaginous tissue of the IVD induces minor susceptibility to scoliosis

AIS occurs in otherwise healthy children during adolescence without associated congenital patterning defects. However, the window of susceptibility to AIS is not defined. Therefore, we set out to empirically determine the susceptible window of AIS initiation using a conditional mouse genetics approach. The ATC mouse strain allows us to test whether *Adgrg6* has a temporal function in committed cartilaginous tissues in spine alignment. We utilized two strategies for induction of recombination: (i) embryonic induction from E0.5–P20 and (ii) perinatal induction from P1–P20. To determine the knockdown efficiency of ADGRG6 in ATC; *Adgrg6*$^{f/f}$ mutant mice, we performed IHC analysis (*Figure 6—figure supplement 1*) and FISH analysis (*Liu et al., 2019*) on spine sections of both control and ATC; *Adgrg6*$^{f/f}$ mutants confirmed that ADGRG6/*Adgrg6* expression was effectively reduced in the cartilaginous endplate and growth plate of ATC; *Adgrg6*$^{f/f}$ mice using either embryonic or perinatal induction strategies. Interestingly, we consistently observed retention of ADGRG6 expression in some regions of the inner annulus fibrosis of the ATC; *Adgrg6*$^{f/f}$ mice induced from P1–P20 (*Figure 6—figure supplement 1D, D'*) compared with the mutants induced from P0.5–P20 (*Figure 6—figure supplement 1B, B'*), suggesting that embryonic induction leads to complete recombination in cartilaginous tissues of the IVD.

Longitudinal X-ray analyses demonstrated that both embryonic and perinatal ablation of *Adgrg6* in committed cartilage tissues of the axial skeleton led to moderate to low penetrance of scoliosis in ATC; *Adgrg6*$^{f/f}$ mice (*Figure 6*). Ablation of *Adgrg6* during embryonic development resulted in 25% (*n* = 16) and 16.7% (*n* = 12) of ATC; *Adgrg6*$^{f/f}$ mice showing scoliosis at P20 and P180, respectively (*Figure 6A–C, H*). Cobb angle measurements revealed curve severity between 11° and 43° in scoliotic mutant mice (*Figure 6H*). Ablation of *Adgrg6* during perinatal development in ATC; *Adgrg6*$^{f/f}$ mice showed no evidence of scoliosis when assayed at P20 (0.0%; *n* = 8), while a single ATC; *Adgrg6*$^{f/f}$ mutant mouse (12.5%; *n* = 8) showed late-onset scoliosis when assayed at P180 with a curve of 23.8° (*Figure 6D–F, J*). None of the Cre (-) littermate control mice showed spine deformity at P20 or P180 (*Figure 6G, I*). Our published histological analyses of the ATC; *Adgrg6*$^{f/f}$ mutant spine showed altered ECM component expression and endplate-oriented disc herniations during later stages of adult development (8 months) (*Liu et al., 2019*). Altogether, this suggested that loss of *Adgrg6* in committed cartilages of the axial skeletal is sufficient to generate a mild susceptibility to scoliosis, although with a much lower penetrance compared with removal of *Adgrg6* in the whole spine (i.e., *Col2a1-Cre; Adgrg6*$^{f/f}$) (*Figure 2L*). These results suggest that the pathogenesis of AIS is the result of tissue-level defects, which accumulate during embryonic development.

To test whether this effect was due to a real biological effect or an idiosyncratic property of the ATC strain, we utilized an alternative *Col2a1-CreER*$^{T2}$ strain to target cartilaginous tissues of the spine postnatally (*Chen et al., 2007*). Next, we performed two temporal induction Tamoxifen strategies: (i) perinatal induction from P1–P5 and (ii) postnatal induction from P14–P18. β-Galactosidase staining of spine sections of *Col2a1-CreER*$^{T2}$; *Rosa26*-lsl-*lacZ* mice at P28 showed that perinatal induction (P1–P5) led to near-complete recombination in cells of the cartilaginous endplate, vertebral growth plate, and most cells of the inner annulus fibrosus but not the outer annulus fibrosus (*Figure 6—figure supplement 2A, A'*), similar to the recombination profile of ATC (*Figure 1I–J'*), except that *Col2a1-CreER*$^{T2}$ did not target the nucleus pulposus (*Figure 6—figure supplement 2A*). Unfortunately, postnatal induction (P14–P18) in *Col2a1-CreER*$^{T2}$; *Rosa26*-lsl-*lacZ* mice only led to partial recombination within endplate and growth plate, and some cells of the inner annulus fibrosus, while the outer annulus fibrosus and nucleus pulposus were not targeted (*Figure 6—figure supplement 2B, B'*). We did not observe scoliosis in *Col2a1-CreER*$^{T2}$; *Adgrg6*$^{f/f}$ mutant mice after perinatal induction (P1–P5) at P30 (0.0%; *n* = 8) (*Figure 6—figure supplement 2C, C', J*). However, a single (14.3%; *n* = 7) *Col2a1-CreER*$^{T2}$; *Adgrg6*$^{f/f}$ mice presented with very mild, late-onset scoliosis (12.4°) at P90 (*Figure 6—figure supplement 2D', J*). The lower penetrance of scoliosis in *Col2a1-CreER*$^{T2}$; *Adgrg6*$^{f/f}$ mice was comparable with the results observed in the ATC; *Adgrg6*$^{f/f}$ mice induced from P1–P20 (*Figure 6J*). We also found no evidence of scoliosis in *Col2a1-CreER*$^{T2}$; *Adgrg6*$^{f/f}$ mice induced at P14–P18 at either P30 or P90 (0.0%; *n* = 5 for *Col2a1-CreER*$^{T2}$; *Adgrg6*$^{f/f}$ and 0.0%; *n* = 4 for controls, respectively) (*Figure 6—figure supplement 2F–H, K, L*). Altogether, our results confirmed that *Adgrg6* plays only a minor role in the postnatal regulation of cartilaginous tissues for maintaining spine alignment in the mouse. However, we showed a slight increase in the incidence of

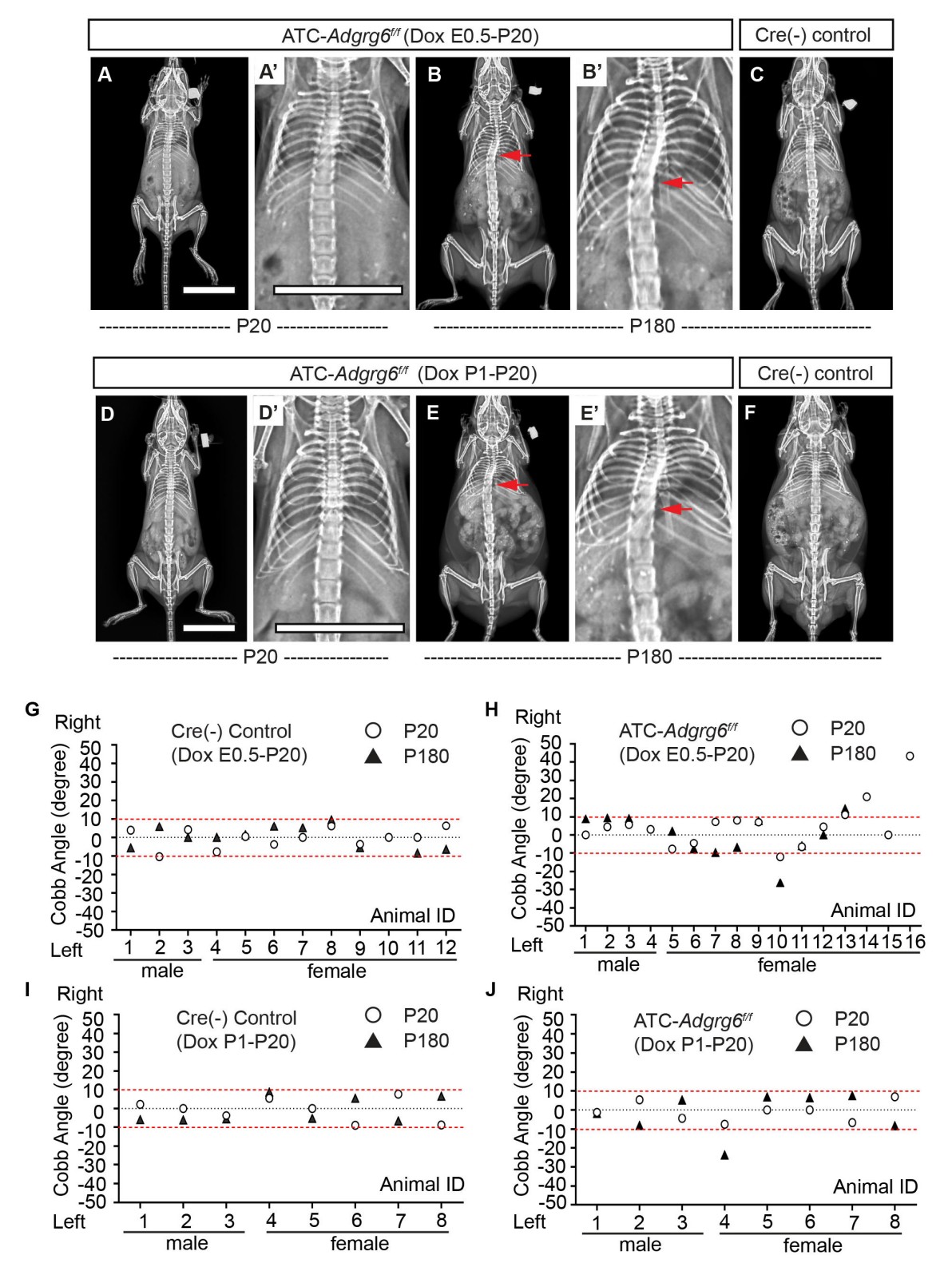

**Figure 6.** Ablation of *Adgrg6* in cartilaginous tissues leads to scoliosis in the mouse. (A–F) Representative X-ray images of Cre (-) control and ATC; *Adgrg6^{f/f}* mutant mice at P20 and P180. ATC; *Adgrg6^{f/f}* mice (Dox induction from E0.5–P20) analyzed at P20 and P180 are shown in (A, A') and (B, B'), respectively. ATC; *Adgrg6^{f/f}* mice (Dox induction from P1–P20) were analyzed at P20 (D, D') and P180 (E, E'). Corresponding Cre (-) control mice analyzed at P180 is shown in (C) and (F). Scoliosis is indicated with red arrows in (B, B') and (E, E'). (G–J) Longitudinal analyses of Cobb angle values of

*Figure 6 continued*

Cre (-) control mice and ATC; *Adgrg6^{f/f}* mice at P20 and P180. For embryonic induction (E0.5–P20), *n* = 12 mice for Cre (-) controls at P20 and P120; *n* = 16 and 12 for ATC; *Adgrg6^{f/f}* mice at P20 and P180, respectively. For perinatal induction (P1–P20), *n* = 8 mice for Cre (-) control at P20 and P180; *n* = 8 for ATC; *Adgrg6^{f/f}* mice at P20 and P180. Thresholds of scoliosis (Cobb angle >10°) are indicated with two red dot lines. Scale bars: 10 mm. The online version of this article includes the following source data and figure supplement(s) for figure 6:

**Source data 1.** Cobb angle measurements of mice with *Adgrg6* ablation in cartilaginous tissues.
**Figure supplement 1.** ADGRG6 expression is dramatically reduced in ATC; *Adgrg6^{f/f}* mice.
**Figure supplement 2.** Loss of *Adgrg6* in cartilaginous tissues leads to low penetrance of scoliosis.

scoliosis in the ATC; *Adgrg6^{f/f}* mice with embryonic deletion of *Adgrg6* compared to perinatal deletion. These results further support a model where the contributions of cartilaginous elements for spine alignment are established during embryonic development in the mouse.

## Ablation of *Adgrg6* in dense connective tissues leads to increased susceptibility to late-onset scoliosis

Connective tissue disorders such as Ehlers–Danlos and Marfan syndromes display scoliosis as a common symptom (*Hoffjan, 2012*) and generalized joint hypermobility is a risk factor for developing AIS (*Haller et al., 2018*). To address a specific role of *Adgrg6* in dense connective tissues of the spine, we generated *Scx-Cre; Adgrg6^{f/f}* mutant mice, which were indistinguishable from littermate controls at birth. Longitudinal X-ray analyses revealed that 55.6% (*n* = 9) of *Scx-Cre; Adgrg6^{f/f}* mice exhibited scoliosis at P120, with curve severity ranged between 12° and 30° (*Figure 7B, B', F*). Interestingly, none of these mice showed scoliosis at P40 (*n* = 9) (*Figure 7A, A' , F*). No scoliosis was observed in Cre (-) or Cre (+) littermate control mice at either P40 or P120 (*Figure 7C–E*). Histological analysis revealed some mildly wedged IVDs and shifted nucleus pulposus within the scoliotic curve of the *Scx-Cre; Adgrg6^{f/f}* mutant spine at P120 (*Figure 7—figure supplement 1C*), with IVDs outside of the curve region displaying typical morphology (*Figure 7—figure supplement 1A, B*). Altogether, this demonstrates that dense connective tissues of the spine are essential for maintaining spine alignment later during adult development (after P40) and may act synergistically with the IVD to promote typical spine alignment during postnatal development in the mouse.

## Ablation of *Adrgrg6* compromises the biomechanical properties of tendons

Next, to determine if the loss of *Adgrg6* alters the mechanical properties of dense connective tissues, we used tendon fascicles isolated from tails of 12-week-old *Scx-Cre; Adgrg6^{f/f}* mice and Cre (-) controls as a proxy for paraspinal ligaments. Tail tendons originate from the same *Scx*-expressing somatic progenitor population in the syndetome as axial tendons and ligaments (*Murchison et al., 2007*). Moreover, the fascicle represents the tissue-level 'functional unit' of tendons and can easily be extracted as an individual, intact unit for highly reproducible characterization of structure-function relationships (*Wunderli et al., 2020*). We examined each fascicle to measure the external diameter, and the cross-sectional area was calculated based on these measurements (*Figure 7G, H*). We found that the total cross-sectional area was increased in *Scx-Cre; Adgrg6^{f/f}* tendons compared to Cre (-) controls (p=0.0037) (*Figure 7G–I*). Micromechanical tensile testing revealed that deletion of *Adgrg6* resulted in mechanically weaker tendons with reduced elastic moduli (p=0.0395) and a trend of reduced failure stresses (p=0.0589) (*Figure 7K, N*). We observed no differences in failure force or failure strain (*Figure 7L, M*). This data suggests that ablation of *Adgrg6* in *Scx*-expressing cells compromises the biomechanical properties of dense connective tissues, which likely contributes to the pathogenesis of late-onset scoliosis in *Scx-Cre; Adgrg6^{f/f}* mice. Altogether, this study demonstrates the essential role of ADGRG6 and its tissue-level function within the cartilaginous tissues and dense connective tissue of the spine and highlights its role in the maintenance of homeostatic signaling and biomechanical properties of these spine elements to regulate postnatal spine alignment in the mouse.

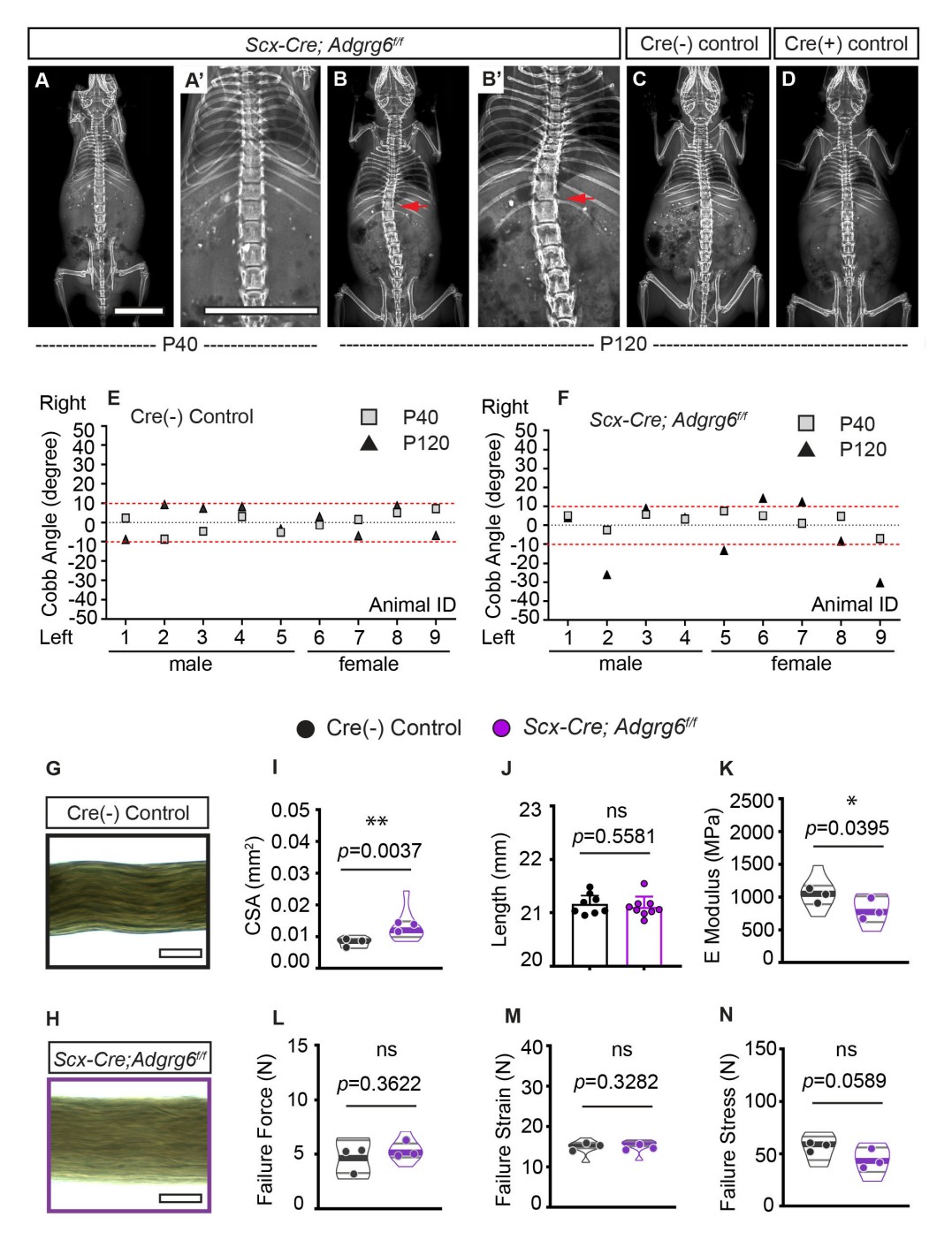

**Figure 7.** Ablation of *Adgrg6* in dense connective tissues leads to late-onset scoliosis and compromised biomechanical properties of the tendons. (A–D) Representative X-ray images of Cre (-) control and *Scx-Cre; Adgrg6^{f/f}* mutant mice. *Scx-Cre; Adgrg6^{f/f}* mice were analyzed at P40 (A, A') and P120 (B, B'). Cre (-) control and Cre (+) control mice analyzed at P120 are shown in (C) and (D), respectively. Scoliosis is indicated with red arrows in (B, B'). (E, F) Longitudinal analyses of Cobb angle values of Cre (-) control mice and *Scx-Cre; Adgrg6^{f/f}* mice at P40 and P120. Cre (-) control mice, *n* = 9 mice at both P40 and P120; *Scx-Cre; Adgrg6^{f/f}* mutant mice, *n* = 9 at both P40 and P120. Thresholds of scoliosis (Cobb angle >10°) are indicated with two red dot lines. (G–N) Biomechanical characterization of Cre (-) control and *Scx-Cre; Adgrg6^{f/f}* mutant tendons at 12 weeks. Representative phase-contrast images of tail fascicles isolated from Cre (-) control and *Scx-Cre; Adgrg6^{f/f}* mutant mice are shown in (G) and (H). Quantification of fascicles' cross-sectional area (CSA) (I), initial length (J), elastic modulus (E Modulus) (K), failure force (L), failure strain (M), and failure stress (N) are also shown. *n* = 8–9 fascicles isolated from three mice. For (I) and (K–N), *n* = 8–9 fascicles (violin plots) isolated from three different mice (close circles). For (J), *n* = 8 and 9 fascicles isolated from Cre (-) control mice or *Scx-Cre; Adgrg6^{f/f}* mutant mice, respectively. Bars are plotted with mean and 95% CI. Cross-sectional area (I) (mean: Cre [-] control: 0.008 [95% CI 0.0069, 0.0095]; *Scx-Cre; Adgrg6^{f/f}*: 0.013 [95% CI 0.0094, 0.0169]) and elastic modulus (K) (mean: Cre [-] control:

*Figure 7 continued on next page*

*Figure 7 continued*

1060 [95% CI 863.8, 1256]; *Scx-Cre; Adgrg6^{f/f}*: 810.9 [95% CI 644.6, 977.3]) were significantly different between control and mutant groups. The statistical difference is evaluated by unpaired *t*-test with Welch's correction for (**J, K, L, N**), and with Mann–Whitney test for non-normally distributed data (**I, M**). The p-value for each comparison is shown. ns: not significant. Scale bars: 10 mm in (**A, A'**); 100 µm in (**G, H**).

The online version of this article includes the following source data and figure supplement(s) for figure 7:

**Source data 1.** Cobb angle measurements and biomechanical testing of mice with *Adgrg6* ablation in dense connective tissues.

**Figure supplement 1.** *Scx-Cre; Adgrg6^{f/f}* mice display mildly wedged intervertebral discs (IVDs) within the curve.

## Discussion

AIS is the most common pediatric disorder worldwide, yet defining the underlying causes remains a persistent challenge for pediatric medicine. A wide range of hypotheses have been proposed to explain its origins, including deformities in skeletal structural elements, neurological abnormalities, and altered biomechanical loading of the spine (*Cheng et al., 2015*; *Latalski et al., 2017*; *Newton Ede and Jones, 2016*). There are multiple animal models proposed for understanding the pathogenesis of AIS, with differing levels of validity (*Liu and Gray, 2018*). We suggest that *Col2a1-Cre; Adgrg6^{f/f}* mutant mice are a unique genetic model of AIS, with construct validity (*Willner, 1984*). In that, these mice models both reflect the natural history of AIS (this work) and are derived by disruption of a gene associated with AIS in humans (*Kou et al., 2013*; *Kou et al., 2018*). Next, we demonstrate that alterations in *Adgrg6* signaling, in part through cAMP/CREB signaling, specifically in dense connective tissues, with a minor contribution from the cartilaginous tissues of the IVD, are key drivers of scoliosis. Finally, our results suggest that stimulation of cAMP/CREB signaling may provide a pathway for therapeutic interventions of AIS.

There is a good agreement for the involvement of cartilaginous tissues in the etiology of AIS. For instance, patients with AIS frequently exhibit wedging of IVDs associated with shifted nucleus pulposus to the convex side of the curve (*Little et al., 2016*); this pathology is precisely recapitulated in *Col2a1-Cre; Adgrg6^{f/f}* mutant mice. Furthermore, T2-weighted magnetic resonance imaging signal distributions in the IVDs can discriminate severity of AIS (*Gervais et al., 2012*), and vertebral growth plate abnormalities are frequently found to locate near the apex of the curve (*Day et al., 2008*), suggesting that alterations of the IVD may proceed the initiation of scoliosis. In agreement, AIS patients exhibit IVD degeneration, including disorganization of chondrocytes in the vertebral growth plate (*Day et al., 2008*), reduced proteoglycan contents in the endplate and nucleus pulposus (*Shu and Melrose, 2018*; *Urban et al., 2001*), and altered distribution of collagen fibers in the annulus fibrosus (*Akhtar et al., 2005*). Some patients also exhibit an increased incidence of Schmorl's nodes (endplate-oriented disc herniations) (*Buttermann and Mullin, 2008*). Interestingly, we found similar endplate defects in both *Col2a1-Cre; Adgrg6^{f/f}* and ATC; *Adgrg6^{f/f}* mice mutant mice (*Liu et al., 2019*), suggesting that *Adgrg6* is vital for the physiology of the cartilaginous endplate. By cartilage-specific conditional ablation of *Adgrg6* during embryonic development, we showed a mild increase in the susceptibility to scoliosis in mouse. Altogether, these findings support a model that the regulation of cartilaginous tissues of spine plays a crucial role during initiation of scoliosis.

An interesting question remains as to how *Adgrg6* regulates the homeostasis of cartilaginous tissues. Previously we showed that *Adgrg6* regulates IVD homeostasis in part by negative regulation of STAT3 signaling (*Liu et al., 2019*). Here, we extend our analysis of *Adgrg6* signaling, demonstrating an additional role for *Adgrg6* in cAMP/CREB signaling in the IVD and in tendon. cAMP signaling has been shown to have chondroprotective effects in cartilage, stimulating expression of several anabolic markers, such as *Col2a1* and *Acan* (*Kosher et al., 1986*; *Malemud et al., 1986*), and can interrupt matrix metalloproteinase-mediated cartilage degeneration in articular cartilage (*Karsdal et al., 2007*). Moreover, cAMP signaling is known to positively regulate SOX9, a master regulator for both development and maintenance of cartilaginous tissues of the spine (*Akiyama et al., 2002*; *Henry et al., 2012*). cAMP signaling stimulates phosphorylation of SOX9 protein, which enhances its DNA binding affinity and transcriptional activity (*Huang et al., 2001*; *Huang et al., 2000*). Moreover, CREB has also been shown to bind to the *SOX9* promoter and induce its expression (*Kanazawa et al., 2014*; *Piera-Velazquez et al., 2007*). In agreement, we show that stimulation of cAMP by addition of FSK can activate pCREB in the absence of *Adgrg6* function and partially restore *Sox9* expression in ATDC5 cell culture. In the future, the use of targeted injections of cAMP/CREB

stimulating drugs or genetically encoded constructs will address the efficacy of this pathway as a modifying therapeutic in the preclinical mouse models of AIS presented here.

The onset of idiopathic scoliosis usually occurs during adolescence in humans (*Cheng et al., 2015*). *Adgrg6* appears to be largely dispensable for processes crucial for overall spine patterning and formation of the IVDs (*Liu et al., 2019*). Here, we showed that ablation of *Adgrg6* in the IVD during embryonic development increased the penetrance of scoliosis in mice. Therefore, our observations suggest that embryonic deletion of *Adgrg6* leads to subtle structural or biomechanical changes of these structural elements of the spine, which in turn may lead to increased susceptibility of scoliosis during periods of rapid skeletal growth. The heritability of scoliosis is not likely explained solely by variants in the *ADGRG6* locus as most variants only explain between 1 and 2% of heritability given the published odd ratios each variant (*Kou et al., 2019*; *Kou et al., 2013*; *Kou et al., 2018*; *Xu et al., 2019*). By identifying downstream effectors of cAMP/CREB signaling in dense connective tissues and the IVD, we may uncover new susceptibility loci involved in AIS. For example, variants associated with both *CREB5* (*Kou et al., 2019*) and *SOX9* (*Miyake et al., 2013*) loci are imputed by GWAS analysis of AIS patients. It will be interesting to identify additional factors that increase the susceptibility to AIS in the *Col2a1-Cre; Adgrg6^{f/f}* mutant mouse background.

In contrast to a recent report (*Sun et al., 2020*), we found that ADGRG6 has no apparent functional role in bone-forming osteoblasts for skeletal morphology. We observed no spinal malformations nor apparent limb or growth phenotype in either *Blgap-Cre; Adgrg6^{f/f}* mice or *Sp7Cre; Adgrg6^{f/f}* mice (*Figure 5A–C* and *Figure 5—figure supplement 1A–C*). Sun et al. also reported that scoliosis was not present in *Sp7Cre; Adgrg6^{f/f}* mice at P120 (*Sun et al., 2020*). However, in contrast with our findings, Sun et al. reported that deletion of *Adgrg6* (*Gpr126*) with *Sp7-Cre*, but not *Lysm-Cre* (which targets osteoclast) or *Col2a1-Cre*, resulted in a significant reduction in overall body size and the bone length. The authors attributed this effect primarily due to delayed osteoblast differentiation (*Sun et al., 2020*). We and others have shown that *Col2a1-Cre* recombines in early osteochondral progenitor cells that give rise to osteoblasts and chondrocytes (*Liu et al., 2019*; *Long et al., 2001*). On the other hand, *Sp7-Cre* is activated in many cells/tissues in addition to osteoblasts, including but not limited to hypertrophic chondrocytes, adipocytes, vascular smooth muscle, perineural and stromal cells in the bone marrow, olfactory glomerular cell, and intestinal epithelial cells (*Chen et al., 2014*; *Liu et al., 2013*). Therefore, the presence of a phenotype only in the *Osx-Cre* but not *Col2a1-Cre* mutant mice could be due to an unidentified cell type or is the effect of the *Osx-Cre* transgene itself (*Davey et al., 2012*; *Huang and Olsen, 2015*; *Wang et al., 2015*). In summary, while we provide evidence against the role of ADGRG6 in osteoblasts, we suggest that additional studies are warranted to understand these divergent results.

*Scx-Cre; Adgrg6^{f/f}* mutant mice—targeting recombination only in dense connective tissues—displayed a high penetrance of scoliosis comparable with that observed in *Col2a1-Cre; Adgrg6^{f/f}* mice. Connective tissues, including ligamentous structures and tendon insertions of the spine, along with the annulus fibrosus of IVD and paraspinal muscles, play a crucial role in controlling spinal stability (*Bogduk, 2016*). Not surprisingly, defects in connective tissues are reported in a range of scoliosis conditions. For instance, reduced fiber density and nonuniform distribution of elastic fibers were observed in the ligamentous tissues in some AIS patients compared with the healthy individuals (*Hadley-Miller et al., 1994*). Moreover, patients diagnosed with connective tissue disorders such as Marfan's syndrome commonly display scoliosis (*Glard et al., 2008*) and rare variants in the Marfan's associated fibrillin genes *FBN1/2* are associated with AIS in humans (*Buchan et al., 2014*; *Sheng et al., 2019*). Here, we show that *Scx-Cre; Adgrg6^{f/f}* mice exhibit reduced elastic moduli in the tendon fascicles of the tail, suggesting that scoliosis in these *Adgrg6* mutant mouse models of AIS is primarily driven by biomechanical instability of the paraspinal tendons and ligaments. Altogether, these results show that dense connective tissues play a crucial role in late-onset scoliosis. Identifying additional factors regulating the biomechanical properties of dense connective tissues may further inform a common pathogenesis of scoliosis phenotypes observed in both AIS and other connective tissue disorders.

Altogether, this study provides evidence that biomechanical defects in cartilaginous and dense connective tissues due to defective ADGRG6/cAMP/CREB signaling may underlie the pathogenesis of AIS. Furthermore, our studies suggest that stimulation of cAMP in spine tissues may be a fruitful treatment approach to disrupt the initiation, progression, and severity of AIS.

# Materials and methods

## Key resources table

| Reagent type (species) or resource | Designation | Source or reference | Identifiers | Additional information |
|---|---|---|---|---|
| Gene (*Mus musculus*) | *Adgrg6* | GenBank | Gene ID: 215798 | |
| Genetic reagent (*Mus musculus*) | *Adgrg6* conditional knockout | Taconic | Model #: TF0269 | *Adgrg6$^{f/f}$* |
| Genetic reagent (*Mus musculus*) | B6.129S4-*Gt(ROSA) 26Sortm1Sor*/J | Jackson Laboratory | Stock no.: 003474 | *Rosa26*-lsl-*lacZ* |
| Genetic reagent (*Mus musculus*) | ATC | *Dy et al., 2012* | | A gift from Dr. Véronique Lefebvre |
| Genetic reagent (*Mus musculus*) | *Col2a1-Cre* | *Long et al., 2001* | | A gift from Dr. Fanxin Long |
| Genetic reagent (*Mus musculus*) | *Col2a1-CreER$^{T2}$* | *Chen et al., 2007* | | A gift from Dr. Matthew Hilton |
| Genetic reagent (*Mus musculus*) | *Scx-Cre* | *Blitz et al., 2009* | | A gift from Dr. Ronen Schweitzer |
| Genetic reagent (*Mus musculus*) | B6.FVB-Tg(BGLAP-cre)1Clem/J | Jackson Laboratory | Stock no.: 019509 | *Bglap-Cre* |
| Genetic reagent (*Mus musculus*) | B6.Cg-Tg(Sp7-tTA,tetO-EGFP/cre)1Amc/J | Jackson Laboratory | Stock no.: 006361 | *Sp7-Cre* |
| Cell line (*Mus musculus*) | ATDC5 | Sigma | Sigma: 99072806 | |
| Cell line (*Mus musculus*) | *Adgrg6* KO ATDC5 | *Liu et al., 2019* | | CRISPR constructed to knock down the expression of *Adgrg6* in ATDC5 cells |
| Antibody | Anti-GPCR GPR126 (ADGRG6) (rabbit polyclonal) | Abcam | ab117092 | IHC 1:500 |
| Antibody | Anti-SOX9 (rabbit Polyclonal) | EMD Millipore | AB5535 | IHC 1:200 |
| Antibody | Anti-collagen II Ab-2 (clone 2B1.5) (mouse monoclonal) | Thermo Scientific | MS235B | IHC 1:100 |
| Antibody | Recombinant anti-collagen I antibody (rabbit monoclonal) | Abcam | ab138492 | IHC 1:1000 |
| Antibody | Phospho-CREB (Ser133) (87G3) (rabbit monoclonal) | Cell Signaling | #9198 | IHC: 1:200 WB: 1:1000 |
| Antibody | GAPDH (14C10) | Cell Signaling | #2118 | WB: 1:2500 |
| Antibody | IRDye 800CW Goat anti-Rabbit IgG Secondary Antibody | LI-COR | P/N: 926-32211 | WB: 1:10,000 |
| Sequence-based reagent | *b-actin*-F | Sigma | qPCR primers | AGATGTGGATC AGCAAGCAG |
| Sequence-based reagent | *b-actin*-R | Sigma | qPCR primers | GCGCAAGTTAG GTTTTGTCA |

*Continued on next page*

*Continued*

| Reagent type (species) or resource | Designation | Source or reference | Identifiers | Additional information |
|---|---|---|---|---|
| Sequence-based reagent | *Col1a1*-F | Sigma | qPCR primers | GCATGGCCAAG AAGACATCC |
| Sequence-based reagent | *Col1a1*-R | Sigma | qPCR primers | CCTCGGGTT TCCACGTCTC |
| Sequence-based reagent | *Adgrg6*-F | Sigma | qPCR primers | CCAAAGTTGG CAATGAAGGT |
| Sequence-based reagent | *Adgrg6*-R | Sigma | qPCR primers | GCTGGATCAG GTAGGAACCA |
| Sequence-based reagent | *Col10a1*-F | Sigma | qPCR primers | CTTTGTGTGCC TTTCAATCG |
| Sequence-based reagent | *Col10a1*-R | Sigma | qPCR primers | GTGAGGTACA GCCTACCAGTTTT |
| Sequence-based reagent | *Col2a1*-F | Sigma | qPCR primers | ACTGGTAAGTG GGGCAAGAC |
| Sequence-based reagent | *Col2a1*-R | Sigma | qPCR primers | CCACACCAA ATTCCTGTTCA |
| Sequence-based reagent | *Acan*-F | Sigma | qPCR primers | CGTGTTTCCAA GGAAAAGGA |
| Sequence-based reagent | *Acan*-R | Sigma | qPCR primers | TGTGCTGATC AAAGTCCAG |
| Sequence-based reagent | *Sox9*-F | Sigma | qPCR primers | AGGAAGCTGG CAGACCAGTA |
| Sequence-based reagent | *Sox9*-R | Sigma | qPCR primers | CGTTCTTCAC CGACTTCCTC |
| Chemical compound, drug | Forskolin | Sigma | Sigma: F6886 | |
| Software, algorithm | GraphPad Prism software | GraphPad Prism (https://graphpad.com) | | |

## Mouse strains

All animal studies were conducted according to institutional and national animal welfare guidelines and were approved by the Institutional Animal Care and Use Committee at The University of Texas at Austin (protocol AUP-2018-00276). All mouse strains were described previously, including *Adgrg6*$^{f/f}$ (Taconic #TF0269) (**Mogha et al., 2013**), *Rosa26*-lsl-*lacZ* (**Soriano, 1999**); ATC (**Dy et al., 2012**), *Col2a1-Cre* (**Long et al., 2001**), *Col2a1-CreER*$^{T2}$ (**Chen et al., 2007**); *Scx-Cre* (**Blitz et al., 2009**), *Bglap-Cre* (**Zhang et al., 2002**), and *Sp7-Cre* (**Rodda and McMahon, 2006**). All mouse strains were kept on a C57BL/6J (JAX:000664) background. Doxycycline (Dox) was administered to ATC; *Adgrg6*$^{f/f}$ and littermate controls by two strategies: (i) inducing from embryonic day (E) 0.5 to postnatal day (P) 20 by ad libitum feeding of Dox-chow (Test Diet, 1814469) to plugged isolated females, and supplemented via intraperitoneal (IP) injections of the pregnant dames with Dox (Sigma D9891) once/week (10 mg/kg body weight) throughout the pregnancy until the pups were weaned at P20; and (ii) inducing from P1–P20 by ad libitum feeding of Dox-chow to the mothers after the pups were born, and supplemented with IP injections of the mothers with Dox once/week (10 mg/kg body weight) until the pups were weaned at P20. Tamoxifen (Sigma, T5648) was dissolved in filter-sterilized corn oil at a concentration of 20 mg/ml and administered to *Col2a1-CreER*$^{T2}$; *Adgrg6*$^{f/f}$ and Cre (-) littermate controls by two strategies: (i) inducing from P1–P5 via injection into the milk dot of the pups once/day for five consecutive days (20 μl at P1, 30 μl at P2 and P3, and 40 μl at P4 and P5); and (ii) inducing from P14–P18 via IP injections once/day (1 mg/10 g) for five consecutive days. ATC; *Rosa26*-lsl-*lacZ*, *Col2a1-CreER*$^{T2}$; *Rosa26*-lsl-*lacZ*, and the corresponding Cre (-) littermate controls were induced with the same strategies.

## Analyses of mice

Radiographs of mouse skeleton were generated using a Kubtec DIGIMUS X-ray system (Kubtec T0081B) with auto-exposure under 25 kV. Cobb angle was measured on high-resolution X-ray images with the software Surgimap (https://www.surgimap.com), as previously described (*Cobb, 1958*).

Histological analysis was performed on thoracic (T5–T12) spines fixed in 10% neutral-buffered formalin for 3 days at room temperature, followed by 2–5 days of decalcification in Formic Acid Bone Decalcifier (Immunocal, StatLab). After decalcification, bones were embedded in paraffin and sectioned at 5 µm thickness. Alcian Blue Hematoxylin/Orange G (ABH/OG) and Safranin O/Fast Green (SO/FG) staining was performed following standard protocols (Center for Musculoskeletal Research, University of Rochester). Immunohistochemical analyses were performed on paraffin sections with traditional antigen retrieval (Collagen II: 4 mg/ml pepsin in 0.01 N HCl solution, 37°C water bath for 10 min; SOX9 and Collagen I: 10 µg/ml proteinase K in 1× PBS, room temperature for 10 min; ADGRG6 and pCREB: 10 mM Tris-EDTA with 0.05% Triton-X-100, pH 9.0, 75°C water bath for 5 min) and colorimetric developed development methodologies with the following primary antibodies: anti-SOX9 (EMD Millipore AB5535, 1:200), anti-Collagen II (Thermo Scientific MS235B, 1:100), anti-GPCR ADGRG6 (Abcam, ab117092, 1:500), anti-collagen I (Abcam, ab138492, 1:1000), and anti-phospho-CREB (Ser133) (Cell Signaling 87G3, #9198, 1:200). The β-galactosidase staining was performed on frozen sections as previously described with modifications (*Liu et al., 2015*). Briefly, mouse spines or P1 pups were harvested and fixed in 10% neutral-buffered formalin for 2 hr or overnight at 4°C, respectively, and thoroughly washed in 1× PBS. Spines were decalcified with 14% EDTA at 4°C for 1 week or overnight, washed in a sucrose gradient, embedded with Tissue-Tek OCT medium, snap-frozen in liquid nitrogen, and sectioned at 10 µm with a Thermo Scientific HM 550 cryostat. Both spine sections and P1 pups were stained with X-gal (0.5 mg/ml) at 37°C overnight protected from light. P1 pups were cleared in 50% glycerol (in 1× PBS and 1% KOH) for 2 weeks at room temperature with gentle rocking and stored in 80% glycerol.

MicroCT analysis was performed on thoracic regions of the spine (T5–T12) in control and *Bglap-Cre; Adgrg6^{f/f}* mutant mice at P120. The spines were fixed in 10% neutral-buffered formalin for 3 days at room temperature, thoroughly washed, and scanned on an Xradia at 100 kV at 9.65 µm true voxel resolution at the University of Texas High-Resolution X-ray Computed Tomography Facility (http://www.ctlab.geo.utexas.edu).

## GAG analysis

8–10 pieces of IVD and adjacent vertebral growth plates were isolate from three Cre (-) control mice and two *Col2a1-Cre; Adgrg6^{f/f}* mutant mice at P20, and snap-frozen in liquid nitrogen. The samples were then ground by pestle in Pronase buffer (0.1 M Tris-HCl, pH 8.0, containing 2 mM CaCl$_2$ and 1% Triton X-100), homogenized by sonication (Barnstead probe homogenizer) for 30 s, and suspended in 2.5 ml of Pronase buffer. Samples were digested with Pronase (0.8 mg/ml) for 24 hr at 50°C with gentle shaking. After 24 hr, a second aliquot of 1.6 mg Pronase was added, and the samples were digested for another 24 hr, followed by incubation at 100°C for 15 min to inactivate the reaction. The samples were then adjusted with 2 mM MgCl$_2$ and 100 mU benzonase, incubated at 37°C for 2 hr, and centrifuged at 4000 g for 1 hr. The supernatant was applied to a DEAE-Sepharose-microcolumn, washed with loading buffer (0.1 M Tris-HCl, pH 8.0) and washing buffer (pH acetate buffer), and then eluted in 2 M ammonium acetate. The acetate salt was removed via lyophilization and the sample was reconstituted in deionized water. These isolated GAG materials were further digested with either Chondroitinase ABC or Hyaluronidase from *Streptomyces hyalurolyticus* at 37°C for over 24 hr, followed by incubation at 100°C for 5 min to inactivate the enzymes. Samples were centrifuged at 14,000 rpm for 30 min before introduction to the HPLC.

SAX-HPLC was carried out on an Agilent system using a 4.6 × 250 mm Waters Spherisorb analytical column with 5 µm particle size at 25°C. A gradient from low to high salt was used to elute the GAG disaccharides. The flow rate was 1.0 ml/min, and the injection of each sample was 10 µl for the positive controls and samples. Detection was performed by post-column derivatization and fluorescence detection. Commercial standard disaccharides (Dextra Laboratories) were used for the identification of each disaccharide based on elution time, as well as calibration. A positive control digestion of HA was used for Hyaluronidase digestions.

## Mechanical testing

Three 12-week-old Cre (-) control mice and *Scx-Cre; Adgrg6^f/f* mutant mice were euthanized and stored in a −20C° freezer until needed. On the day of mechanical testing, tails were thawed and dissected from the body using surgical blades. Fascicles were extracted by holding the tail from its posterior end with surgical clamps and gently pulling off the skin until bundles of fascicles were exposed. Each fascicle was examined under a microscope (Motic AE2000, 20× magnification) for visible signs of damage, and for measuring the external diameter. Specimens with frayed ends, visible kinks, or diameters less than 90 μm were discarded. Micromechanical tensile testing was performed using a custom-made horizontal uniaxial test device to generate load-displacement and load-to-failure curves (10 N load cell, Lorenz Messtechnik GmbH, Germany). Briefly, 2 cm fascicle specimens were carefully mounted and kept hydrated in PBS, as previously described (*Fessel and Snedeker, 2011*). Each fascicle underwent the following protocol: pre-loading to 0.015 N (L0: 0% strain), five cycles of precondition to 1% L0, an additional 1% strain cycle to calculate the tangential elastic modulus, and then ramped to failure to 20% strain under a predetermined displacement rate of 1 mm/s. The load-displacement data were processed using a custom-written MATLAB script (Matlab R2018b, v.9.5.0.944444, MathWorks, Inc). Tangent elastic moduli were calculated from the linear region of stress-strain curves (0.5–1%). Nominal stress was estimated based on the initial cross-sectional area. The cross-sectional area was calculated from microscopic images assuming fascicles have a perfect cylindrical shape.

## Cell culture

Verified WT ATDC5 cells were purchased from Sigma (Sigma, 99072806). The *Adgrg6* KO ATDC5 cell line was generated in our lab as previously described (*Liu et al., 2019*). Cells have been authenticated from ATCC via STR profiling, with no human and/or African green monkey loci been detected. Cells from each thawed cryovial were monitored and confirmed negative for mycoplasma contamination using the MycoAlert PLUS Mycoplasma Detection Kit (LONZA, LT07-701). Both WT ATDC5 cells and *Adgrg6* KO ATDC5 cells were maintained in DMEM/F-12 (1:1) medium (Gibco, 11330032) supplemented with 5% FBS and 1% penicillin/streptomycin. Cells were cultured in 24-well cell culture plate and matured in DMEM/F-12 (1:1) medium supplemented with 5% FBS, 1% penicillin/streptomycin, 1% ITS premix (Corning, 354352), 50 μg/ml ascorbic acid, 10 nM dexamethasone, and 10 ng/ml TGF-β3 (Sigma, SRP6552) for 7 days, with stimulation of 2 μM FSK (Sigma F6886) or DMSO control. After 7 days, cells were processed with RNA isolation and real-time RT-PCR.

## RNA isolation and real-time RT-PCR

To confirm the reduction of *Adgrg6* expression in osteoblast lineages, we isolated total RNA from fresh frozen, pulverized long bones at P120. Total RNA was isolated using an RNeasy mini kit (Qiagen 74104). Reverse transcription was performed using 500 ng total RNA with an iScript cDNA synthesis kit (Bio-Rad). Real-time RT-PCR (qPCR) analyses were performed as previously described (*Liu et al., 2015*). Gene expression was normalized to b-actin mRNA and relative expression was calculated using the 2-(ΔΔCt) method. The following primers were used to check gene expression: b-actin: AGATGTGGATCAGCAAGCAG/ GCGCAAGTTAGGTTTTGTCA; *Col1a1*: GCATGGCCAAGAAGACATCC/CCTCGGGTTTCCACGTCTC; *Adgrg6*: CCAAAGTTGGCAATGAAGGT/GCTGGATCAGGTAGGAACCA; *Col10a1*: CTTTGTGTGCCTTTCAATCG/GTGAGGTACAGCCTACCAGTTTT; *Col2a1*: ACTGGTAAGTGGGGCAAGAC/CCACACCAAATTCCTGTTCA; *Acan*: CGTGTTTCCAAGGAAAAGGA/TGTGCTGATCAAAGTCCAG; and *Sox9*: AGGAAGCTGGCAGACCAGTA/CGTTCTTCACCGACTTCCTC. Real-time RT-PCR efficiency was optimized and melting curve analyses of products were performed to ensure reaction specificity.

## Western blotting

Both WT and *Adgrg6* KO ATDC5 cells were treated with 10 μM or 20 μM FSK for 30 min before protein extraction. Total proteins were extracted from cells with the M-PER Mammalian Protein Extraction Reagent (Sigma, GE28-9412-79) supplemented with protease and phosphatase inhibitors (Roche). 10 mg of protein from each sample was resolved by 4–15% SDS-polyacrylamide gel electrophoresis and transferred to the nitrocellulose membranes. Western blots were then blocked with LI-COR blocking buffer and incubated overnight with primary antibodies anti-pCREB (Ser133) (Cell

Signaling 87G3, #9198, 1:1000) and anti-GAPDH (Cell Signaling, #2118, 1:2500) at 4°C with gentle rocking. The next day western blots were detected with the LI-COR Odyssey infrared imaging system.

## Statistics

Statistical analyses were performed in GraphPad Prism 9.1.2 (GraphPad Software Inc, San Diego, CA). For Cobb angle and gene expression analysis to compare two or more experimental groups, two-tailed Student's $t$-test, one-way ANOVA, or two-way ANOVA followed by Turkey's multiple comparison test was applied as appropriate. For mechanical testing, the normality of distribution was checked using the Shapiro–Wilk test. Unpaired Student's $t$-test with Welch's correction or Mann–Whitney U-test were used for two-group comparison of parametric and non-parametric data, respectively. A p-value of less than 0.05 is considered statistically significant. Bar graphs were generated with mean and 95% confidence intervals (95% CI), and an exact p-value was reported for each comparison.

## Acknowledgements

We thank Drs. Matthew Harris and John Wallingford for helpful comments before publication. We thank Dr. Fanxin Long for sharing *Col2a1-Cre* mice, Dr. Véronique Lefebvre for sharing ATC mice, Dr. Matthew Hilton for sharing *Col2a1-CreER^T2* mice, and Dr. Ronen Schweitzer for sharing *Scx-Cre* mice. We thank Dr. Parastoo Azadi and Ms. Stephanie Archer-Hartmann at the Complex Carbohydrate Research Center for performing the GAG analysis. This study was supported by the National Institute of Arthritis and Musculoskeletal and Skin Diseases of the National Institutes of Health under Award Number R01AR072009 (RSG), R01AR071967, and R01AR076325 (CMK), and K99AR077090 and F32AR073648 (ZL). It was also supported in part by the National Institutes of Health-funded Research Resource for Integrated Glycotechnology (NIH Award Number 5P41GM10339024) to Parastoo Azadi at the Complex Carbohydrate Research Center, and the Cariplo Foundation (2016-0481), the Vontobel Foundation, and institutional funding of both the ETH Zurich and the University Hospital Balgrist to (JGS).

## Additional information

### Funding

| Funder | Grant reference number | Author |
| --- | --- | --- |
| NIAMS | R01AR072009 | Ryan S Gray |
| NIAMS | R01AR071967 | Courtney M Karner |
| NIAMS | R01AR076325 | Courtney M Karner |
| NIAMS | F32AR073648 | Zhaoyang Liu |
| Vontobel-Stiftung | | Jess G Snedeker |
| NIAMS | K99AR077090 | Zhaoyang Liu |
| Cariplo Foundation | 2016–0481 | Jess G Snedeker |

The funders had no role in study design, data collection and interpretation, or the decision to submit the work for publication.

### Author contributions

Zhaoyang Liu, Data curation, Formal analysis, Funding acquisition, Validation, Investigation, Visualization, Methodology, Writing - original draft, Writing - review and editing; Amro A Hussien, Formal analysis, Investigation, Visualization, Methodology, Writing - review and editing; Yunjia Wang, Formal analysis; Terry Heckmann, Roberto Gonzalez, Data curation; Courtney M Karner, Data curation, Formal analysis, Investigation, Writing - review and editing; Jess G Snedeker, Resources, Supervision, Funding acquisition, Investigation, Writing - review and editing; Ryan S Gray, Conceptualization,

Resources, Data curation, Supervision, Funding acquisition, Investigation, Methodology, Project administration, Writing - review and editing

### Author ORCIDs
Zhaoyang Liu (ID) https://orcid.org/0000-0002-8032-1167
Amro A Hussien (ID) http://orcid.org/0000-0002-9324-9360
Courtney M Karner (ID) http://orcid.org/0000-0003-0387-4486
Ryan S Gray (ID) https://orcid.org/0000-0001-9668-6497

### Ethics
Animal experimentation: This study was performed in strict accordance with the recommendations in the Guide for the Care and Use of Laboratory Animals of the National Institutes of Health. All of the animals were handled according to approved institutional animal care and use committee (IACUC) protocols (AUP-2018-00276) of the University of Texas at Austin.

### Decision letter and Author response
Decision letter https://doi.org/10.7554/eLife.67781.sa1
Author response https://doi.org/10.7554/eLife.67781.sa2

## Additional files

### Supplementary files
• Supplementary file 1. Chondroitin sulfate digestion profile and total hyaluronan content in Cre (-) control and *Col2a1-Cre; Adgrg6*$^{f/f}$ mutant mice at P20. Data shows µg estimated for total reaction volume; *w/w percentage*. CS: chondroitin sulfate; HA: hyaluronan; GAG: glycosaminoglycan; ND: not detected.

• Transparent reporting form

### Data availability
All data generated or analyzed during this study are included in the manuscript and supporting files.

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
