## [Decision Letter]

**Acceptance summary:**

Genetic association studies implicated mutations in ADGR6 in adolescent idiopathic scoliosis. In the manuscript by Liu et al., the authors investigate the role of Adgr6 in spine development in mice and dissect tissue specific contributions leading to late onset scoliosis in knockouts. Furthermore, they implicate Adgr6 in regulating gene expression and the mechanical properties of dense connective tissues via cAMP signaling that are linked to the development of scoliosis.

**Decision letter after peer review:**

Thank you for submitting your article "A G protein-coupled receptor is required in cartilaginous and dense connective tissues to maintain spine alignment" for consideration by *eLife*. Your article has been reviewed by 3 peer reviewers, including Michel Bagnat as the Reviewing Editor and Reviewer #1, and the evaluation has been overseen by Marianne Bronner as the Senior Editor. The following individual involved in review of your submission has agreed to reveal their identity: Matthew Harris (Reviewer #2).

Overall, the reviewers agree in that this work is of high quality and that it should be of interest for the field. The main point (1) that needs revision is the organization of data. In the current order the flow is interrupted, making it difficult to follow. Restructuring of the Results section to achieve a better logical order and better integration between and within the sections is essential to make the findings accessible and of interest to a broad readership.

There are additional points to be considered and those may be addressed with existing data not included in the original manuscript, editorial changes, addition of discussion points or, if appropriate and feasible, by additional data points.

Essential revisions:

1. The data on Figure 4 seems a bit out of place. It would be better to move to the beginning of the paper as it reports the expression of Adgr6 and the activity of the Cre lines used in the study.

2. In Figure 3 the authors show a phenotypic rescue in isolated cells using forskolin. Have the authors attempted to rescue the scoliosis phenotype by injecting forskolin into the peri-spinal tissues of KO mice? If this was unsuccessful or not technically feasible it should be mentioned in the discussion as a similar strategy is proposed to be of therapeutical interest.

3. If possible, a more detailed time course of Adgr6 expression should be added to determine if the pattern changes over time and how this relates to the observed phenotypes.

4. It would be interesting to see if there is early indication for abnormality before the onset of scoliosis by showing histology and gene expression in cartilage, tendon and IVD of P10 mice.

---

## [Author Response]

Essential revisions:1. The data on Figure 4 seems a bit out of place. It would be better to move to the beginning of the paper as it reports the expression of Adgr6 and the activity of the Cre lines used in the study.

Thank you for this constructive advice, we moved the Cre recombination figure from Figure 4 to Figure 1 to help with the flow of the article as suggested. Subsequently the number of other figures has been changed in order to accommodate this move.

2. In Figure 3 the authors show a phenotypic rescue in isolated cells using forskolin. Have the authors attempted to rescue the scoliosis phenotype by injecting forskolin into the peri-spinal tissues of KO mice? If this was unsuccessful or not technically feasible it should be mentioned in the discussion as a similar strategy is proposed to be of therapeutical interest.

Thank you for this suggestion, we are certainly gearing up for this type of approach. However, given our significant restrictions during COVID, we have not been able to refine our mouse protocol for targeted Forskolin injections. Moreover, our current longitudinal study in *Col2a1-Cre; Adgrg6^f/f^* mutant mice indicate that a minimum of 16 mice per genotype will give us 80% power (α<0.05) to obtain statistically significant results for treatment of scoliosis. Thus, we will not be able to carry out this type of experiment during the revision.

Moreover, given the challenges of targeted injection in the spine, we are planning to use an alternative genetic approach to show prove of principle given our lack of expertise with drug studies. For this we will be using a lox-STOP-Gs-coupled DREEAD mouse strain that will allow for tissue specific stimulation of Gs/cAMP in the presence of an exogenous ligand CNO, which is well-established in other contexts in mice. We aim for this approach be complemented by more refined, localized, possibly slow release of Forskolin in the thoracic spine, working in close collaboration with a pharmacology group with expertise in drug studies in mice.

We have pulled back own our claims that stimulation of cAMP may be therapeutic until we can provide better evidence of its effectiveness in our mouse model.

3. If possible, a more detailed time course of Adgr6 expression should be added to determine if the pattern changes over time and how this relates to the observed phenotypes.

Thank you for this suggestion. We added IHC analyses of ADGRG6 expression in control mice at different time points: P10 (Figure 1—figure supplement 1 B-B’’), P20 (Figure 1—figure supplement 1 D) and P180 (Figure 6—figure supplement 1 A-A’’ and C-C’’). We observed similar expression pattern of ADGRG6 in cartilaginous tissues of the IVD along the course of postnatal development, though with lower expression level as observed at P1 (Figure 1—figure supplement 1 A-A’’). We also added IHC analyses of *Col2a1-Cre; Adgrg6^f/f^* mice at P20 (Figure 1—figure supplement 1E) and quantified the percentile of ADGRG6 (+) cells in different compartments of the IVD (Figure 1—figure supplement 1F). Overall we observed reduced ADGRG6 expression in the entire IVD in the mutant mice, especially in the scoliotic individuals (red dots, Figure 1—figure supplement 1F). We also observed more reduced ADGRG6 expression in the annulus fibrosus of the convex side of the spine compared with that in the concave side in the scoliotic *Col2a1-Cre; Adgrg6^f/f^* mice (Figure 1—figure supplement 1E). We have updated this information in both main text and the figure legends.

4. It would be interesting to see if there is early indication for abnormality before the onset of scoliosis by showing histology and gene expression in cartilage, tendon and IVD of P10 mice.

Thank you for this suggestion. We collected control and *Col2a1-Cre; Adgrg6^f/f^* mutant mice (n=4 for each group) at P10 and performed histological staining and IHC analyses (Figure 3—figure supplement 1 E-L). We do not observe overt histopathological changes in neither the spine nor IVD tissues prior to the onset of scoliosis (Figure 3—figure supplement 1 E-F’). IHC analyses reveal reduced expression of pCREB in the nucleus pulposus of the mutant mice prior to the onset of scoliosis (Figure 3—figure supplement 1K), however the expression of SOX9 was not obviously changed at this time point (Figure 3—figure supplement 1L). This data confirms that *Adgrg6* is largely indispensable for normal spine patterning and development, and as a potential regulator of SOX9, alteration of cAMP/CREB signaling emerges prior to the onset of scoliosis in the *Col2a1-Cre; Adgrg6^f/f^* mutant mice. We have updated this information in both the main text and the figure legends.